

# Modeling hydropower operations at the scale of a power grid: a demand-based approach

Laure Baratgin[1,2], Jan Polcher[1], Patrice Dumas[3], and Philippe Quirion[2]

[1]LMD/IPSL, École Polytechnique, Institut Polytechnique de Paris, ENS, PSL Research University, Sorbonne Université, CNRS, Palaiseau France
[2]CIRED, AgroParisTech, CNRS, Ecole des Ponts Paris Tech, CIRAD, EHESS, Nogent sur Marne Cedex, France
[3]CIRAD, UMR CIRED, F-34398 Montpellier, France

**Correspondence:** Laure Baratgin (laure.baratgin@lmd.ipsl.fr)

**Abstract.** Climate change and evolving water management practices may have a profound impact on hydropower generation. While hydrological models have been widely used to assess these effects, they often present some limitations. A major challenge lies in the modeling of release decisions for hydropower reservoirs, which result from intricate trade-offs, involving power sector dispatch, competing water uses and the spatial allocation of power generation within the grid.

5   To address this gap, this study introduces a novel demand-based approach for integrating hydropower within the routing module of land surface models. First, hydropower infrastructures are placed in coherence with the hydrological network and links are built between hydropower plants and their supplying reservoirs to explicitly represent water transfers built for hydropower generation. Then, coordinated dam operation is simulated by distributing a prescribed electric demand to be satisfied by hydropower over the different power plants on the power grid, while considering the operational constraints associated with
10  the multipurpose nature of most dams.

To validate our approach, this framework is implemented within the water transport scheme of a land surface model and assessed with the case study of the French electrical system. We drive the model with a high-resolution atmospheric reanalysis and prescribe the observed national hydropower production as the total power demand to be met by hydropower infrastructures. By comparing the simulated evolution of the stock in reservoirs to the observations, we find that the model simulates realistic
15  operations of reservoirs and successfully satisfies hydropower production demands over the entire period. We highlight the roles of uncertainties in estimated precipitation and of the limited knowledge of hydropower infrastructure on the estimation of production. Finally, we show that such an integration of hydropower operations in the model improves the simulations of river discharges in mountainous catchments affected by hydropower.





## 1 Introduction

### 1.1 Background and motivation

Hydroelectric power is set to play a pivotal role in many of the world's power grids in the coming decades, providing low-carbon and dispatchable generation capacity. The International Energy Agency (IEA) projects a 17% increase in installed hydroelectric capacity over the 2021-2030 period (IEA, 2021). Power grids relying on hydropower production are, however, subject to the unpredictability of weather and climate. Consequently, assessing the potential impact of drought events or climate change on hydropower production is a major concern for building resilient energy systems. Numerous studies (e.g., (Lehner et al., 2005; Van Vliet et al., 2016; Turner et al., 2017; Zhou et al., 2018; Voisin et al., 2020)) reveal significant effects of climate change in different geographic regions, including southwestern Europe and France.

As emphasized in the methodological review conducted by Turner and Voisin (2022), these studies commonly employ global hydrological models (GHMs) or land surface models (LSMs) driven by atmospheric projections generated by global climate models (GCMs). They simulate the regional-scale hydrological cycle, offering gridded assessments of surface runoff and streamflow, which are subsequently used to derive hydropower production estimates.

However, this estimation process from streamflow to hydropower production is challenging for three main reasons.

Firstly, water can be stored in reservoirs for future use. The timing of the releases is the result of the optimization of power grid management and the coordinated operation of plants located in different water catchments. Representing in climate models these intricate economic and spatial trade-offs, which drive the operation of hydroelectric reservoirs, is complex.

Secondly, reservoirs that feed hydropower plants are often multi-purposes and are also operated to satisfy other water uses, namely irrigation or tourism.

Thirdly, hydropower production can involve inter-catchment water transfers, particularly prevalent in mountainous regions where water is stored at higher elevations before being channeled to power plants located in the valleys. Representing these short-scale processes within regional models poses further complications.

Existing studies adopt diverse strategies to represent these complex operation of hydroelectric reservoirs, which are generally categorized into two main approaches (Nazemi and Wheater, 2015b).

On the one hand, simulation algorithms rely on predefined rules to compute daily or monthly reservoirs releases. These rules are most often a function of reservoir inflow and filling level, inspired by the pioneering work of Hanasaki et al. (2006). Such rules generally do not take the specific purpose of the reservoir into account, except for irrigation-aimed reservoirs, the release rules of which also depend on downstream irrigative demands. Hydropower and flood control reservoir are therefore represented identically, including in models such as MOSART-WM, a reservoir scheme used for many recent studies (Zhou et al., 2018; Voisin et al., 2020; Ralston Fonseca et al., 2021). However, Abeshu et al. (2023) demonstrate that, in practice, the operation of hydropower reservoirs differs from that of flood control reservoirs. One significant distinction is that hydropower reservoirs are operated with the intention of maximizing water level in the reservoir at the moment of release, leveraging a high



head, while flood control reservoirs are drained preemptively to avoid potential flood events. The realism of the operations simulated in models that do not distinguish hydroelectric reservoirs can therefore be questioned.

Alternatively, the rules can be defined based on target curves, which set daily target water levels and determine releases accordingly. This approach is utilized, for example, in Vic-Res (Dang et al., 2020), which is employed in some hydropower studies (Chowdhury et al., 2021; Siala et al., 2021). This method accounts for the seasonal behavior of hydropower reservoirs, such as drawdown during drier months and recharge during the monsoon season, but it misses the representation of short-term operational constraints.

On the other hand, optimization algorithms based on the pioneering work of Haddeland et al. (2006) determine the optimal release for each dam. The objective function to optimize varies depending on the primary purpose of the reservoir, aiming to maximize individual production for hydropower reservoirs. One limitation of this approach is the requirement of future inflow knowledge for each reservoir over the optimization horizon, necessitating multiple runs of the model to optimize reservoir releases.

More recent models utilize stochastic dynamic programming (SDP) approaches to account for the uncertain nature of reservoir inflows (Turner et al., 2017). In these model, monthly releases are determined at each time step to maximize the total expected benefit from hydropower, considering both immediate and future benefits. However, these methods consider each reservoir independently and often employ large time steps (monthly) to reduce computational strain.

When the models distinguish the different usages of reservoir, they classified the reservoirs based only on their primary purpose (Abeshu et al., 2023), which does not allow for the representation of all the constraints applying to most of hydroelectric reservoirs, that are often dedicated to other purposes as well.

Moreover, none of these studies operate the dams as a network that takes advantage of the spatial complementarity of climatic regions or cascading effects.


   Finally, to our knowledge, none of these large-scale studies explicitly model the water transfers from reservoirs to power plants. In most cases, they employ the flow rate within the grid cell corresponding to the power plant to deduce its production, without considering the actual reservoir location (Van Vliet et al., 2016; Zhou et al., 2018; Voisin et al., 2020). However, this approach can potentially lead to an overestimation of production, given that the flow at the plant site is necessarily greater than

that at the upstream dam site.

## 1.2   Objectives

The objective of our study is to present the original methodology we developed to estimate hydropower production at the scale of a regional power grid, based on the simulations of a GHM or LSM, answering the three challenges previously identified: (i) considering the coordinated management of the entire power system at the scale of the regional grid; (ii) accounting for

multi-purposes objective of reservoirs that store water for hydropower production; (iii) representing the inter-catchment water transfers from reservoirs to power plants.





Our approach is inspired by demand-based irrigation reservoir management models pioneered by Hanasaki et al. (2006). In these algorithms, a demand point (irrigated area) is connected to a supply point (river), with the water demand of the downstream irrigated area driving upstream reservoir releases (Nazemi and Wheater, 2015b). Zhou et al. (2021) developed such a

module to operate irrigative reservoirs in the ORCHIDEE (ORganizing Carbon and Hydrology In Dynamic EcosystEm) land surface model (Krinner et al., 2005) framework, which was validated for the Yellow River basin. In these algorithms, the irrigative demand input is inferred based on withdrawal observations or model estimations.

In our methodology, hydropower plants are linked to reservoirs whose release depends on the demand for hydropower pro-
duction addressed to the plant. This "hydropower demand" derives from power dispatch choices made at the grid level.
We assume that this hydropower demand results from the decisions of a social planner in charge of the dispatch of the power demand to the various available electricity sources. The social planner knows the potentials and costs of all the units available in the network area, as well as the electricity demand and calls the appropriate source when the notional price of electricity corresponds to the unit opportunity cost. We do not explicitly represent this side of the electricity system decisions, but as-
sume corresponding "hydropower demand" to drive the operations of hydropower reservoirs in our model. At the scale of the power grid, the balance between power demand and generation is the primary concern, regardless of the specific locations of consumption and production sites. Therefore we consider an aggregated hydropower demand that needs to then be allocated to the different plants within the grid.

We implement our proposed methodology in the ORCHIDEE LSM, but it aims to be usable in all LSMs and GHMs. The first steps of building a river network that represent inter-catchment hydropower transfers and defining rules for reservoir releases are generic and only require basic information on dams and plants locations and on hydro-electric power plant capacities. To validate the effectiveness of the approach, we apply it to the French power grid, which heavily relies on hydropower, accounting for approximately 10% of its production. A calibration step is added, which requires more information on individual plants
to adjust the efficiencies of the power plants. Finally, simulated and actual operations of hydropower reservoirs are compared.

The paper is structured as follows: Sect. 2 describes the proposed methodology and its originality. Sect. 3 introduces the data and methods used for our case study of the French power grid and assesses the performance of ORCHIDEE in reproducing river discharges over this area. Sect. 4 estimates and discusses the production biases of the model in the case study and presents
the calibration method we use to address them. Sect. 5 details the modeling results, and finally Sect. 6 discusses these results and concludes by outlining future perspectives of research.



## 2    Presentation of the modeling approach

Our method relies on three main novelties: building a river network that includes most hydropower-related infrastructures and represents inter-basin hydropower transfers (Sect. 2.1), implementing a reservoir scheme that accounts for multi-purpose reservoirs (Sect. 2.2), and using hydropower demand to infer hydropower reservoir operations (Sect. 2.3).

### 2.1    Definition of a routing network that includes hydropower connections

#### 2.1.1    General principles

ORCHIDEE is a LSM, initially designed to be coupled to an atmospheric global circulation model (Krinner et al., 2005). In this study, we use it in stand-alone mode, forced by an atmospheric forcing dataset.

The typical spatial resolution of LSMs is imposed by the atmospheric grid of the forcing files, generally from 0.5° (around 50km) for large-scale implementations to 0.1° (around 10km) for regional implementations. However, human activities such as irrigation or urban areas operate at much higher spatial resolution, typically within a few kilometers.

The concept of hydrological transfer units (HTUs) has been introduced in routing modules to bridge the gap between such differences in atmospheric and hydrological processes resolutions and provide the opportunity to incorporate human activities in such models (Nguyen-Quang et al. (2018) for ORCHIDEE model). HTUs correspond to sub-grid river basins which allow the runoff generated in one atmospheric grid cell to flow into multiple neighboring atmospheric grid cells. These smaller units allow for a better representation of the river system and its interaction with human activities including hydropower.

Three types of hydropower plants are distinguished, with different implications on locations:

- Run-of-river plants lack any storage capacity and generate electricity accordingly to the instantaneous river discharge at the plant location. There is no difficulty involved in the location of the plants;

- Reservoir plants are fed by reservoirs which can store a specified water volume and are often also used for other purposes, which may constrain the operations of the plant. Electricity production does not necessarily take place at the location of water storage, therefore the plant and the reservoir need to be located separately;

- Pumped-hydro-storage (PHS) plants are able to pump water from a downstream reservoir to an upstream one during low electric demand periods. The links with both downstream and upstream reservoirs needs to be determined.

As an example of different locations of reservoir and power plant, the power plant of "La Bathie" - the largest reservoir power plant in France - uses water from Roselend reservoir, which is located about 20 km away. At a kilometric resolution, this implies horizontal water transfers between these two locations (water withdrawal and restitution) that cannot be merged, as has been done in previous studies (Zhou et al., 2018). This requires the reconstruction of the hydroelectric water supply network within the routing network of ORCHIDEE.




We proceed in three steps as illustrated in Fig. 1.

First (Fig. 1-b), we place dams and hydropower plants on a high-resolution river network (MERIT (Yamazaki et al., 2019) is
used in this study), based on geo-referenced and upstream area provided in the databases. The location procedure is detailed in
Appendix A and the infrastructures datasets used for our study of France are presented in Appendix B.

Then, we build the adduction network by identifying supposed connections between power plants and dams that feed them (see
Sect. 2.1.2 for more details on the procedure to build the adduction network).

Finally (Fig. 1-c), we form HTUs by aggregating MERIT pixels in an atmospheric grid cell with the same general flow direction
following the procedures described in Nguyen-Quang et al. (2018) and Polcher et al. (2023).

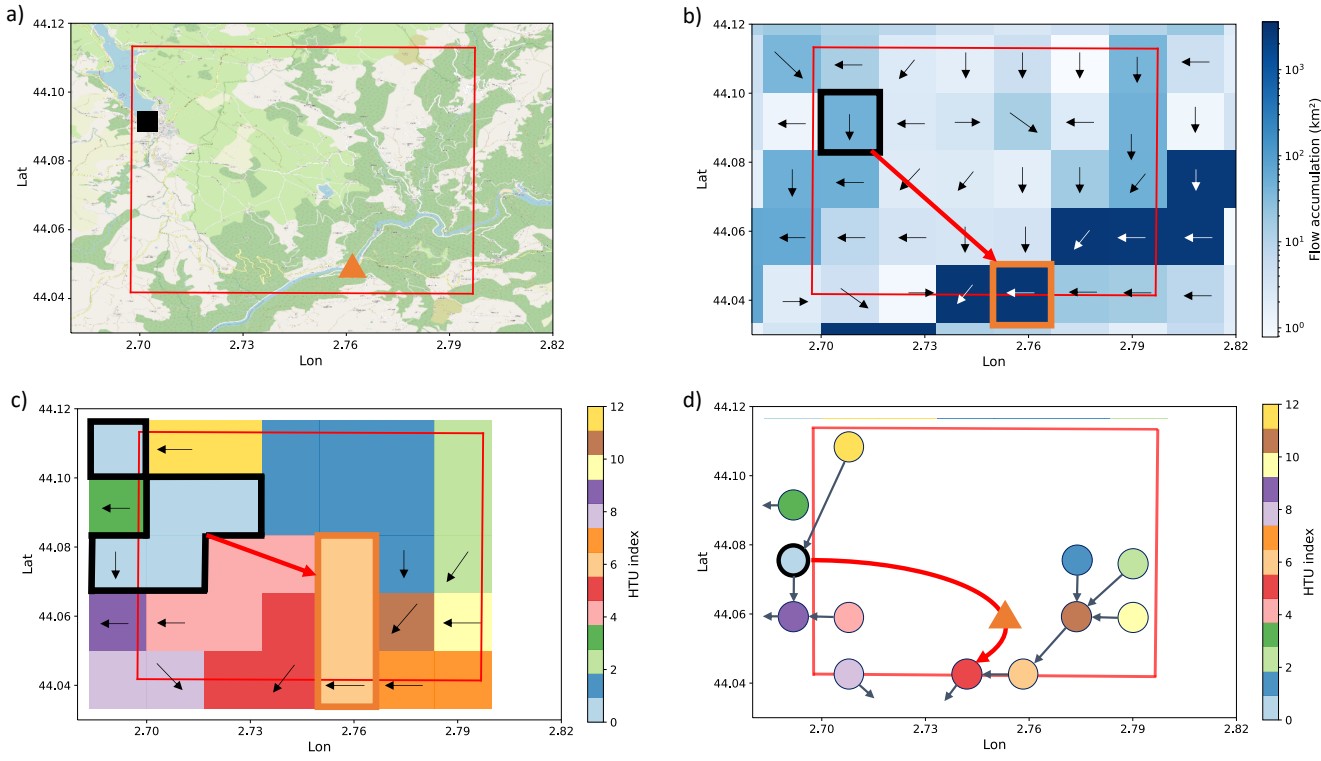

**Figure 1.** Illustration of the procedure to build ORCHIDEE routing network with the example of Pouget hydropower plant in France. (a)
Geographic context of Pouget power plant (orange triangle) and its feeding reservoir (black square indicating the location of the dam). Red
grid indicates the atmospheric grid. (b) Flow directions and accumulation for the MERIT pixels overlapping the atmospheric grid. MERIT
pixels in which we located the power plant and the dam are respectively indicated in orange and black while the red arrows represent the
adduction network link we identify. (c) Resulting HTUs decomposition. The location of the infrastructures is reported in the corresponding
HTUs (d) Corresponding HTUs graph. The HTU containing the dam is indicated with a bold black outline while the power plant (orange
triangle) is placed on the edge between the reservoir and the HTU downstream from the one in which it has been located.





This aggregation results in an HTU network representing natural and human-made water flows. It can be seen as a directional graph (Fig. 1-d) where vertices correspond to HTUs and edges represent directional water flows (natural and human-made for hydropower purposes). Considering this graph, hydropower plants are placed on the edges connecting the HTU of their withdrawal point and the HTU downstream of the one in which they are located. Fig. 2 introduces the notation that will be used throughout the article to index HTU and edges in such graphs. It shows that the water used to produce electricity can follow a different path to the natural flow out of the reservoir. This approach allows for the representation of this distinction independently of the atmospheric resolution.

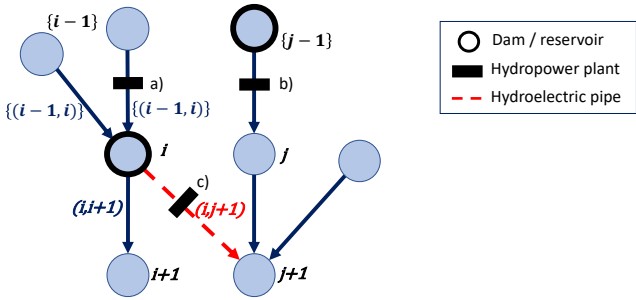

**Figure 2.** Graph representation of the river routing network built.

Each vertex represents an HTU. HTUs containing a dam are represented by bold dark circles. Edges represent existing water flows directions (blue edges for natural water flows and dashed red ones for hydroelectric pipes). Power plants are placed on edges whose water flows they can use to produce power ((a): run-of-river plant, (b)-(c) reservoir plants)).

The indexing convention is also presented on the graph, with integers used for vertices and couples of integers for edges. $i+1$ is the HTU directly downstream of $i$ (natural flow) while $\{i-1\}$ denotes the ensemble of HTU flowing into HTU $i$. Similarly $(i,i+1)$ is the natural outflow edge from HTU $i$ while $\{(i-1,i)\}$ represent the ensemble of inflow edges into HTU $i$, including basin transfers.

### 2.1.2 Adduction network

Reservoir and PHS plants produce power from water releases from upper reservoirs. To explicitly represent this adduction network in our model, we have to identify such connections between a feeding reservoir and a power plant. For each reservoir or PHS plant, we thus select as feeding reservoir the one that maximizes the potential function $\phi = \frac{U*V*h}{d}$, where $U$ is the upstream area of the dam, $V$ is the storage capacity of the reservoir, $h$ is the elevation difference between the plant and the reservoir and $d$ is the horizontal distance between them. Similarly, a downstream reservoir is selected for each PHS plant by maximizing $\phi' = \frac{U*V*(-h)}{d}$. The definition of these potential functions is inspired by similar works aiming to connect an irrigative area to a water supply point (Neverre, 2015; Zhou et al., 2021).

This position algorithm relies on the assumption that each plant is fed by only one reservoir. This assumption is however debatable, especially for plants in mountain areas that may be connected to several reservoirs. In this case, our choice of potential function $\phi$ privileges the reservoir with the largest upstream area since it is likely to determine the production potential of the plants. During calibration (see Sect. 4), plants for which the identification of a single reservoir conducts to a significant





misrepresentation of the plant's hydropower potential are identified and a correction is made by moving the withdrawal point so that it gathers enough water to ensure the observed production is possible.

### 2.1.3 Aggregation to form the HTUs

Attributes and variables describing reservoir and hydropower characteristics of each HTU $i$ and vertex $(i,j)$ are presented in Table 1.

Note that each vertex and edge can respectively contain only one dam or hydropower plant. If several reservoirs are placed on the same HTU during pixels aggregation, their respective volumes for the different uses are summed. If two plants are placed on the same edge, their installed power and pumping capacity as well as their head are summed only if both plants have the same input point. Otherwise, only the plant with the highest installed capacity is kept. As in other studies (Abeshu et al., 2023), all the reservoir attributes are associated with the HTU of the dam (even if its water surface can be larger than the HTU 185 area and its geometry is different from the HTU geometry).

| | | |
|---|---|---|
| vertex | $V_{tot,i}$ | Total maximum storage capacity of the reservoir located in HTU $i$ ($m^3$) |
| | $V_{elec,i}, V_{recr,i}, V_{irri,i}$ | Maximum storage capacity dedicated to respective water uses (hydropower, recreation and irrigation) of the reservoir located in HTU $i$ ($m^3$) |
| | $H_{dam,i}$ | Height of the dam located in HTU $i$ ($m$) |
| | $\boldsymbol{V_i(t)}$ | **Current total volume in the reservoir located in HTU $i$ ($m^3$)** |
| | $V_{min,i}(t)$ | Minimal water volume in the reservoir, it evolves with time to account for recreation uses (see Fig. 5) ($m^3$) |
| | $h_{res,i}(t)$ | Water level in the reservoir ($m$) |
| | $A_{res,i}(t)$ | Surface of the reservoir ($m^2$) |
| edge | $P_{(i,j)}$ | Installed hydropower capacity of the plant located on the edge (MW) |
| | $P'_{(i,j)}$ | Installed pumping capacity of the plant located on the edge (MW) |
| | $H_{(i,j)}$ | Nominal hydraulic head of the plant located on the edge, obtained with a full reservoir ($m$) |
| | $Typ_{(i,j)}$ | Hydropower plant type (run-of-river, reservoir or PHS) |
| | $\eta_{(i,j)}$ | Production efficiency of the plant (conversion of potential energy to power) |
| | $\eta'_{(i,j)}$ | Pumping efficiency of the plant (conversion of power to potential energy) |
| | $E_{(i,j)}(t)$ | Production of the plant on the edge (MWh) |
| | $E'_{(i,j)}(t)$ | Power consumption (MWh) of the plant on the edge associated to water pumping |

**Table 1.** Model attributes and variables describing reservoirs and hydropower. Prognostic variables are distinguished in bold

### 2.2 Dams and reservoir parametrization

In the initial version of ORCHIDEE (Polcher et al., 2023), each HTU $i$ contains three natural water stores, characterized by their time constants (slow aquifer, fast aquifer, and stream storage). To represent water management we add a fourth store to the



HTUs in which dams have been located to represent water storage in the reservoir (Fig. 3). This section presents the continuity
equation for the water volume in this reservoir.

### 2.2.1    Prognostic equations for water stores

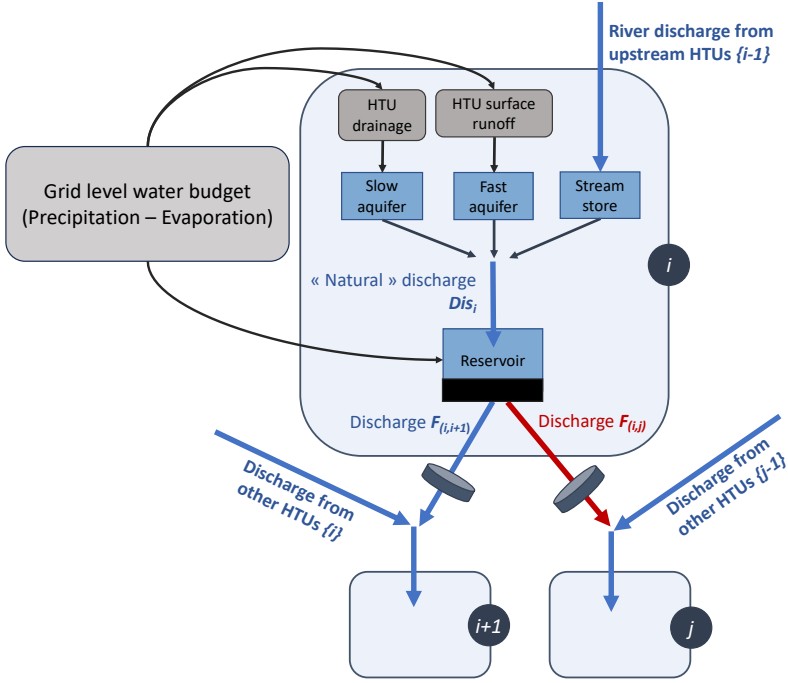

**Figure 3.** Schematic representation of water stores and flows in an HTU $i$

As represented in Fig. 3, the fast aquifer is filled by local runoff generated in the HTU, the slow aquifer by local drainage
generated in the HTU and the stream store by the discharge from upstream HTUs. The equations of these natural water stores
are detailed in previous publications (Zhou et al., 2021; Polcher et al., 2023). They introduce the respective time constants of
the natural stores $g_{stream}$, $g_{fast}$ and $g_{slow}$ (in unit $h.m^{-1}$) and the topographic index calculated for each HTU $\tau_i$ (in unit $m^2$).

The "natural discharge" $Dis_i(t)$ in the HTU $i$ is generated by summing the outflows of the natural water stores (Eq. (1)).
This natural discharge is stored in the reservoir if there is one in the HTU, or routed towards the downstream HTU if there is
not.

$$Dis_i(t) = \frac{1}{\tau_i} * \left( \frac{W_{stream,i}(t)}{g_{stream}} + \frac{W_{fast,i}(t)}{g_{fast}} + \frac{W_{slow,i}(t)}{g_{slow}} \right) \tag{1}$$

Prognostic equation on reservoir volume is then given by:

$$\frac{dV_i}{dt}(t) = Dis_i(t) + p_{res,i}(t) - ev_{res,i}(t) - R_i(t) \tag{2}$$





where $p_{res,i}(t)$ and $ev_{res,i}(t)$ are respectively direct precipitation and evaporation over the reservoir, and $R_i(t)$ is the total water release from the reservoir, which depends on the demands addressed to the reservoir, in the limit of the available capacity. This release is computed based on:

$$R_i(t) = max \left( \min \left( \frac{V_i^\star(t) - V_{min,i}(t)}{g_{res} * \tau_i} \ , \ Dw_i(t) \right), \ \frac{V_i^\star(t) - V_{tot,i}}{g_{res} * \tau_i} \right) \tag{3}$$

where $V_i^\star(t)$ is the theoretical volume to be obtained without any release (Eq. (4)), $V_{min,i}(t)$ is the minimal volume of the reservoir, $g_{res}$ is the time constant of the reservoir, and $Dw_i(t)$ is the total water demand addressed to the reservoir through its outflow edges (Eq. (5), the different water demands are defined in Sect. 2.3).

$$\frac{dV_i^\star}{dt}(t) = Dis_i(t) + p_{res,i}(t) - ev_{res,i}(t) \tag{4}$$

$$Dw_i(t) = \max\left(D_{ecol.,(i+1,i)}(t), D_{irri.,(i+1,i)}(t), D_{elec.,(i+1,i)}(t)\right) + \sum_{j \neq i+1} D_{elec.,(j,i)}(t) \tag{5}$$

Reservoir release $R_i(t)$ generates water flows on the different edges connected to the HTU: $R_i(t) = \sum_j F_{(i,j)}(t)$, where:

$$F_{(i,i+1)}(t) = \min\left(R_i(t), \ \max\left(D_{ecol.,(i+1,i)}(t), D_{irri.,(i+1,i)}(t), D_{elec.,(i+1,i)}(t)\right)\right) + \max\left(R_i(t) - Dw_i(t), 0\right) \tag{6}$$

$$F_{(i,j)}(t) = \min\left(R_i(t) - F_{(i,i+1)}(t) \ , \ D_{elec.,(j,i)}(t)\right), \text{ for } j \neq i+1 \tag{7}$$

The water flow to the river $F_{(i,i+1)}(t)$ is computed before the other flows $F_{(i,j)}(t)$, so we give priority to demands from the river downstream over any demand from power plants not located directly downstream of the reservoir. This is consistent with water management policy in most of the countries, where ecological demand takes priority over other non-vital water uses.

### 2.2.2 Diagnostic variables

**Reservoir water level and surface**

As in previous studies (Fekete et al., 2010; Zhou et al., 2018), we represent each reservoir $i$ in the form of a tetrahedron of height $H_{dam,i}$ and volume $V_{tot,i}$ (Fig. 4).

We checked the validity of this assumption based on data available in the infrastructures dataset (see Appendix B). The height $H$ of a tetrahedron, the area of the opposite face $A$ and the volume $V$ are linked by the relationship $H \times A = 3V$. By regressing $H_{res,i} \times A_{res,i}$ against $V_{res,i}$, we find a slope of 3.28 with a coefficient correlation of 0.93, which validates the geometry hypothesis.

Hence, the relations between the volume $V_i(t)$, the water level $H_{res,i}(t)$ and the area of the reservoir $A_{res,i}(t)$ are given by:

$$H_{res,i}(t) = H_{dam,i} * \left(\frac{V_i(t)}{V_{tot,i}}\right)^{\frac{1}{3}} \tag{8}$$





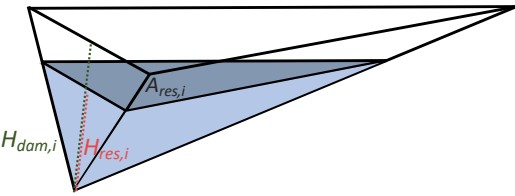

**Figure 4.** Geometry of the reservoir


$$A_{res,i}(t) = \frac{3 * V_i(t)}{H_{res,i}(t)} \tag{9}$$

Direct precipitation and evaporation ($m^3/s$) over the reservoir are then given by $p_{res,i}(t) = P_i(t) * A_{res,i}(t)$ and $ev_{res,i}(t) = Ev_i(t) * A_{res,i}(t)$ where $P_i(t)$ and $Ev_i(t)$ are respectively the precipitation and evaporation over the HTU $i$ (in $m/s$).

### Hydropower production and pumping

The production $E_{(i,j)}(t)$ of an hydropower plant located on the edge $(i,j)$ is:

$$E_{(i,j)}(t) = \min\Big( \rho g \eta_{(i,j)} h_{(i,j)}(t) F_{(i,j)}(t) \, , \, P_{(i,j)} \Big) \tag{10}$$

where $\rho$ is the water density, $g$ is the gravitational constant, $\eta_{(i,j)}$ is the efficiency of the plant (set at 0.9 by default), $h_{(i,j)}(t)$ is the current hydraulic head (which varies with the water level of the reservoir (Eq. (11))) and $P_{(i,j)}$ is the installed capacity

of the power plant.

$$h_{(i,j)}(t) = H_{(i,j)} - (H_{dam,i} - H_{res,i}(t)) \tag{11}$$

     Similarly, the power $E'_{(i,j)}(t)$ consumed by an hydropower plant located on the edge $(i,j)$ to pump $F_{(j,i)}(t)$ is:

$$E'_{(i,j)}(t) = \min\Big( \frac{\rho g h_{(i,j)}(t) F_{(j,i)}(t)}{\eta'_{(i,j)}} \, , \, P'_{(i,j)} \Big) \tag{12}$$

where $\eta'_{(i,j)}$ is the pumping efficiency of the plant (set at 0.85 by default, following literature such as Wessel et al. (2020) and available data of French PHS production and pumping (RTE, b)) and $P'_{(i,j)}$ is the installed pumping capacity of the power plant.

### 2.3   Water demands

Reservoirs store water to fulfill several types of demand, such as domestic and industrial uses, irrigation, energy production (hydropower and thermal plants) or tourism. As this study focuses on hydropower reservoirs, we adopt a simplistic representation of the other water uses and only detail water uses that can constrain hydropower operations.

     In this subsection we describe the modeling approach we adopt to take into account some of these water demands.



### 2.3.1 Ecological demand

In many countries, the environmental laws require a minimum flow $F_{min.,(i,i+1)}$ in the watercourse downstream of a dam $i$, to guarantee the ecological quality of the river. Minimal flow requirements depend on the region. Details for the French study case are presented in Sect. 3.3.1.

Such an ecological demand $D_{ecol.,(j,i)}(t)$ applies to all reservoirs regardless of their intended use:

$$D_{ecol.,(j,i)}(t) = \begin{cases} F_{min.,(i,i+1)}, & \text{if } j = i+1 \\ 0, & \text{else} \end{cases} \tag{13}$$


### 2.3.2 Irrigative demand

Some reservoirs also store water for agriculture. Water withdrawals for irrigation can be made either directly from the reservoir or from the downstream river. Withdrawals from the river require a corresponding release from upstream reservoirs to maintain

low flows.

In this study, the water requirements for irrigation are represented in a highly simplified manner by assuming a need proportional to $F_{min.,(i,i+1)}$ during the summer period. $D_{irri.(i+1,i)}$ is then expressed in Eq. (14). The choice of the proportional factor $\alpha_{irri}$ and the delimitation of the summer period are discussed for our French case study in Sect. 3.3.2.

$$D_{irri.,(j,i)}(t) = \begin{cases} \alpha_{irri} * F_{min.,(i,i+1)}, & \text{if } j = i+1 \ \& \ V_{irri,i} > 0 \ \& \ t \in Summer \\ 0, & \text{else} \end{cases} \tag{14}$$


### 2.3.3 Tourism

In summer, some reservoirs may also become touristic areas where recreational activities are carried out and require the reservoir to be kept at a high level. To ensure proper reservoir filling during the summer season, dam operators follow a filling

guide curve. We define corresponding constraints on $V_{min,i}(t)$ based on previous work and data available for French reservoirs (e.g. François (2013) on the Serre Ponçon reservoir), see Fig. 5.

By default, the minimum volume is set at 10% of the total capacity of the reservoir and is increased to 90% during the touristic period for reservoirs concerned.





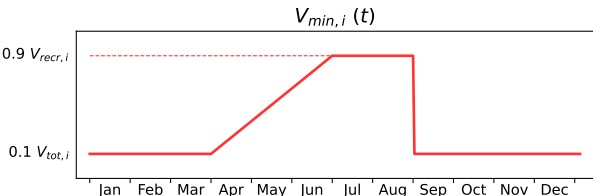

**Figure 5.** Evolution of the minimum volume constraints during the year

### 2.3.4 Hydroelectric demand

Production of hydropower plants is the result of the dispatch of the total power demand among the different power plants on the power grid (Stoft, 2002; Wood et al., 2013). Power generation units are called upon from least to most expensive to meet power demand at a minimal total cost. Run-of-river power plants, whose production is free and non-dispatchable, are called upon first, along with solar and wind power plants, to produce to their maximum potential (as long as it does not exceed demand, otherwise there is a curtailment of their production). On the contrary, the call upon reservoir and PHS power plants is the result

of a much more complex trade-off, aiming to minimize the total power system cost. From the point of view of a social planner, in charge of dispatch decisions and aware of the potentials and costs of all the units available in the network area, as well as the electricity demand, it is thus possible to define a demand for reservoir hydroelectric power generation $D_{res}(t)$, PHS generation $D_{phs}(t)$ and PHS pumping $D'_{phs}(t)$ at each time step. These demands (or production targets) are defined for the whole grid and need then to be distributed among the different plant units to decide the amount of energy generated $E_{(i,j)}(t)$ (and consumed

$E'_{(i,j)}(t)$) at each plant location, that will then drive reservoir release decisions. Indeed, knowing $E_{(i,j)}(t)$, the model deduces the hydroelectric water demand $D_{elec.,(j,i)}(t)$ as the additional water release needed for the plant production (Eq. (15)) and can finally compute the reservoir release based on Eq. (5) and (3).

$$D_{elec.,(j,i)}(t) = \max \left( \frac{E_{(i,j)}(t)}{\rho g \eta_{(i,j)} h_{(i,j)}(t)} - \max \left( D_{ecol.,(j,i)}(t) , D_{irri.,(j,i)}(t) \right) , 0 \right) + \frac{E'_{(j,i)}(t) \eta'_{(j,i)}}{\rho g h_{(j,i)}(t)} \tag{15}$$


To distribute national demand into individual plants production $E_{(i,j)}(t)$ (or consumption for pumping $E'_{(i,j)}(t)$), the model proceeds in two steps.

### 1) Fatal production and pumping:

The model starts by going through all the hydropower plants and calculates the energy they can produce or store without additional release, thanks to spillage (water that overflows) and other releases (ecological or irrigative). Associated production $E_{fatal,(i,j)}(t)$ and energy consumption $E'_{fatal,(i,j)}(t)$) are computed based on Eq. (16) and (17).



$$E_{fatal,(i,j)}(t) = \min\left[ P_{(i,j)} * \frac{h_{(i,j)}(t)}{H_{(i,j)}}, \right.$$

$$\left[ \min\left( \frac{V_i^\star(t) - V_{tot,i}}{g_{res}\tau_i} - \max\left( D_{ecol.,(i+1,i)}(t) , D_{irri.,(i+1,i)}(t) \right), 0 \right) + \right.$$

$$\left. \left. \min\left( \max\left( D_{ecol.,(j,i)}(t) , D_{irri.,(j,i)}(t) \right), \frac{V_i^\star(t) - V_{i,min}(t)}{g_{res}\tau_i} \right) \right] \times \rho g \eta_{(i,j)} h_{(i,j)}(t) \right] \quad (16)$$

$$E'_{fatal,(i,j)}(t) = \min\left[ P'_{(i,j)}, \min\left( \frac{V_j^\star(t) - V_{tot,j}}{g_{res}\tau_i} - \max\left( D_{ecol.,(j+1,j)}(t) , D_{irri.,(j+1,j)}(t) \right), 0 \right) \times \frac{\rho g h_{(i,j)}(t)}{\eta'_{(i,j)}} \right] \quad (17)$$

The remaining production demand to dispatch is then $D_k(t) - \sum_{Typ(i,j)=k} E_{fatal,(i,j)}(t)$ for $k$ in {res,phs,pump}.

**2) Reservoirs withdrawals**: If there is any national production (or pumping) demand left to dispatch, it should be produced by withdrawing water from the reservoirs. In this study, we consider that the reservoirs are used in the decreasing order of their relative filling to produce power, while respecting production constraints (installed capacity of the plant and remaining volume of water in the reservoir). The remaining production is dispatched following this rule, until either all remaining production demand has been satisfied, or no more plants can produce. This rule leads to the equalization of relative filling at the end of the each time step. This is equivalent to implementing a uniform rule curve for all reservoirs, as has been done in Dang et al. (2020). Another advantage offered by this rule is that it leads to a production spread out over the whole territory. All plants are required to produce a little power each day, close to the so-called stable productions modeled in other studies (Sterl et al., 2020).

### 2.4 Validation diagnostics

The performance of our model to estimate hydropower production will be assessed based on three main diagnostics.

#### 2.4.1 Hydraulic stock

Hydraulic stock notion refers to the total energy that can be produced using energy stored in all the reservoirs of the power grid, it is defined by (Eq. (18)).

$$S(t) = \sum_{(i,j)s.a.Typ_{(i,j)}=reservoir} \int_{V_{min,i}(t)}^{V_i(t)} \rho g \eta_{(i,j)} h_{(i,j)}(V) \, dV \quad (18)$$

#### 2.4.2 Time-series of simulated production by hydropower plant type

For a hydropower plant type $k$, the simulated production $E_k(t)$ is given by $E_k(t) = \sum_{(i,j)s.a.Typ_{(i,j)}=k} E_{(i,j)}(t)$ (where $E_{(i,j)}(t)$ is defined in Eq. (10) and $k$ belongs to {run-of-river, reservoir, phs}).



### 2.4.3 Annual hydropower potential (AHP) of an individual plant

For run-of-river and reservoir plants, we define $AHP_{(i,j)}(y)$ as the maximum energy which could be produced by the plant $(i,j)$ over the year $y$ in our simulation.

To compute it, we run a simulation in which the hydroelectric demand $D_{res,t}$ is fixed to infinite, leading all hydroelectric reservoir to release water within the limits of water availability and the installed capacity of the plant. Simulated water flow $F_{i,j}(t)$ at the plant location is then used to compute $AHP_{(i,j)}(y)$ based on Eq. (10), considering the average head of each plant $\overline{h_{(i,j)}}$:

$$AHP_{(i,j)}(y) = \int\limits_{t \ in \ y} \min\Big( \rho g \eta_{(i,j)} \overline{h_{(i,j)}} F_{(i,j)}(t) \ , \ P_{(i,j)} \Big) dt \tag{19}$$

The average head $\overline{h_{(i,j)}}$ is determined based on Eq. (11), taking the average reservoir water level. Observations of the average hydraulic stock combined with an assumption of identical average filling for every reservoir allows to compute the average filling with Eq. (18). This leads to an average filling of 63%. Then, Eq. (11) leads to $\overline{h_{(i,j)}} = H_{(i,j)} - 0.14 * H_{dam,i}$.

## 3 Data and methods for the test case over France

To validate this modeling approach, the proposed method is applied to the French power grid.

### 3.1 ORCHIDEE setup

In this study, ORCHIDEE is run in stand-alone mode, forced with the SAFRAN meteorological data set (Quintana-Segui et al., 2008). SAFRAN (Système d'Analyse Fournissant des Renseignements Atmosphériques à la Neige) is a surface reanalysis resulting from the optimal interpolation between the vertical profiles of the atmosphere derived from ERA-40 atmospheric reanalysis and surface observations. It provides the required atmospheric variables - temperature, relative humidity at 2 m,
wind speed, downward radiation (shortwaves and longwaves) and precipitation (solid and liquid) - at an hourly time step over a 8 × 8 km grid that covers France and upstream part of the international catchments beyond its borders.

To estimate the sensitivity of ORCHIDEE's simulations to the uncertainties of precipitation, we built two alternative atmospheric forcings by replacing precipitation data in SAFRAN with other precipitation datasets: COMEPHORE (Tabary et al.,
2012) and SPAZM (Gottardi et al., 2008). These datasets are presented in detail in Appendix C1 and their relatives differences with SAFRAN are displayed in Fig. C1 .

COMEPHORE dataset provides observations of surface precipitation accumulation over metropolitan France at an hourly and kilometric resolution based on a synthesis of radar and rain gauge data. We build a meteorologic dataset SAF_COM by replacing precipitation data in SAFRAN with data from COMEPHORE. As COMEPHORE does not distinguish solid and
liquid precipitations, we keep SAFRAN's hourly ratio of solid/liquid precipitations when possible and discriminate based on the air temperature otherwise. The differences in annual mean precipitation between SAFRAN and COMEPHORE are



generally small, with an average deviation inferior to 1.0% in COMEPHORE compared to SAFRAN (Fig. C1). However we find a small seasonal bias as this average deviation goes from -2.0% for Winter period to +1.9% in Summer. Moreover, discrepancies increase dramatically in mountainous regions, especially in the Alps and in the Pyrenees. For grid points with an
average elevation above 1000m, the annual mean precipitation in COMEPHORE is, on average, 10.4% lower.

SPAZM is a daily reanalysis of precipitation at the kilometer-scale, developed by EDF, the main electricity producer in France. We interpolate the daily precipitation data from SPAZM to the hourly scale and merge it with SAFRAN data to create the alternative forcing dataset SAF_SPAZM. As for SAF_COM, we keep SAFRAN's hourly ratio of solid/liquid precipitations when possible. Compared to SAFRAN, precipitations are in average 2.7% higher in SPAZM with an average bias of 7.0% in
Summer, against 2.1% in Winter. Bias is heterogeneously spread over France (Fig. C1) with bigger differences on the highest reliefs, without a clear sign (average deviation of +3.9% for grid points above 1000m).

The vegetation distribution map used in ORCHIDEE is derived from the ESA-CCI Land Cover dataset at 0.05° resolution for the year 2010. The soil background albedo map is derived from the MODIS albedo dataset aggregated at 0.5° resolution.
Soil texture distribution maps is obtained from Reynolds map (Reynolds et al., 2000) at 5-arc-min resolution with 12 USDA soil texture classes (at 30-cm depth).

In our study, ORCHIDEE performs the energy and water budgets at a 15 minutes time step and hydropower operations are performed at the same time step.

## 3.2 Locating hydroelectric infrastructures on the routing network

As explained in Sect. 2.1, the first step in building the routing network is to locate the infrastructures on the high-resolution river network based on the information provided in infrastructures datasets. The infrastructures datasets we use in this study are presented in Appendix B. We assess here the quality of this location.

Following the procedure outlined in Fig. 1, we locate the infrastructures on the MERIT river network and construct the HTUs routing graph based on the simplification of this MERIT network (resolution of 2km) on the SAFRAN atmospheric grid (resolution of 8 km). HTUs area can thus theoretically vary from 0 to 64 $km^2$ and the average area of HTUs in our graph is 4.73 $km^2$.

The upstream area of an HTU is defined recursively as the sum of the HTU area and the upstream area of all its tributaries.
For each hydroelectric infrastructure, we compare in Fig. 6 its reference upstream area (from database or MERIT network) to the upstream area of the HTU in which it is located. For most of the structures, the positioning error is lower than 20%. Some dams with a small upstream area are, however, located in HTUs with a higher upstream area, due to resolution constraints.





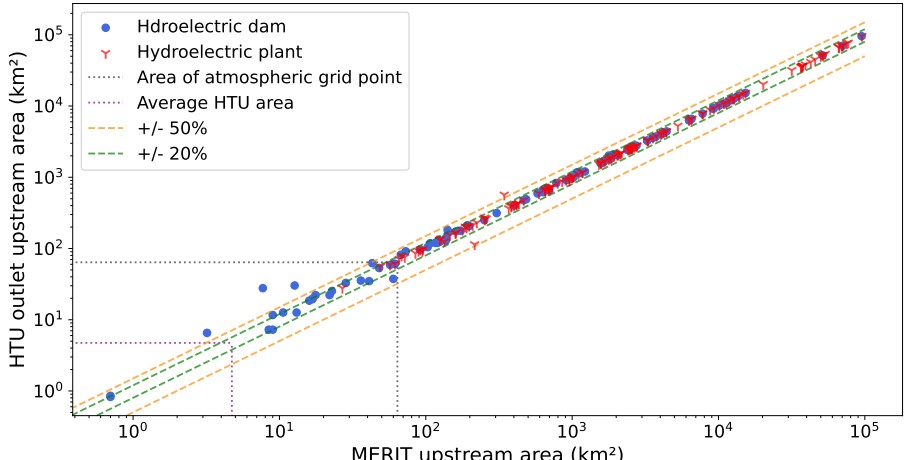

**Figure 6.** Comparison of the initial upstream area of the infrastructure (referenced in the database or upstream area of the MERIT pixel on which it is placed) with its final upstream area in the HTUs graph. Blue dots represent hydroelectric reservoirs (reservoirs that have been associated with power plants during the adduction network building step) and red signs represent hydropower plants. Green and orange dashed lines delineate a respective error of +/- 20% and +/- 50% while grey and purple dotted lines refer to the respective atmospheric grid point area and average area of an HTU.

## 3.3 Data for water demands and validation

### 3.3.1 Ecological and irrigative demands

In France, minimal flow requirements are defined relatively to the mean interannual flow upstream of the dam $\overline{Dis_i}$ (Code de l'Environnement, Article L214-18). They are summarized in Table 2. We ran a twenty-year SAFRAN simulation without reservoir operations to calculate $\overline{Dis_i}$ at dam locations.

| | $\overline{Dis_i} > 80m^3/s$ | $\overline{Dis_i} < 80m^3/s$ |
|---|---|---|
| Dam intended for hydropower purpose | $F_{min,(i,i+1)}(t) = 5\% * \overline{Dis_i}$ or flow immediately upstream of the dam if it is lower | $F_{min,(i,i+1)}(t) = 5\% * \overline{Dis_i}$ or flow immediately upstream of the dam if it is lower |
| Dam intended for other purpose | $F_{min,(i,i+1)}(t) = 5\% * \overline{Dis_i}$ or flow immediately upstream of the dam if it is lower | $F_{min,(i,i+1)}(t) = 10\% * \overline{Dis_i}$ or flow immediately upstream of the dam if it is lower |

**Table 2.** French legal requirements for ecological flow, $\overline{Dis_i}$ is the mean interannual flow downstream of the dam

To account for the irrigative purposes of some reservoirs, we increase the minimal flow requirement downstream of reservoirs intended for irrigation during the summer period (June 1st to September 30th) by setting $\alpha_{irri} = 8$. This choice is based on

information available from French reservoir concession contracts, which sometimes specify the volume of water reserved for





irrigation. In the case of Serre-Ponçon, for example, the concession contract stipulates a reserve of 200 million $m^3$, to be used for irrigation, between July 1 and September 30. If we consider a constant withdrawal spanning three months, this corresponds to a $25m^3/s$ flow, which is 45% of the $55m^3/s$ mean interannual flow at this location, and thus 9 times larger than $F_{min}$, which is set to 5%, as explained above.

### 3.3.2 Hydropower production demand

As this study aims to validate our proposed reservoir operations model, we take historic time-series of production and consumption as hydropower demand prescribed to the model. We can thus assess if the reservoir operations performed by the model when it is forced by historical atmospheric dataset can meet the observed production.

Data of observed production for hydropower plants in the French power grid are published from 2015 by the French electricity transmission system operator RTE at a 30-minute timestep for 4 categories (RTE, a):

- **Run-of-river production** that gathers the production of pure run-of-river power plants and reservoir power plants with a limited storage capacity called "poundage" (distinction defined by French operators)

- **Reservoir production** that gathers the production of reservoir power plants with a greater storage capacity

- **PHS production** that gathers the production of PHS power plants

- **PHS pumping** that gathers the consumption of PHS power plants for pumping

### 3.3.3 Validation data

In France, hydroelectricity is produced by companies that do not share precise data on the production of their power plants or the filling of the reservoirs they manage. Similarly, discharge data from gauging stations near hydroelectric power plants are often inaccessible to the public. This limits the available data for validating our model.

However, as a delegate of public services, RTE provides data, often aggregated at the national level, which allows us to calibrate and validate our model as shown in the following two sections.

The available data is:

- National time-series of production and consumption by hydroelectric sector (run-of-river, lakes and PHS) at 30-minute time step from 2015 (RTE, a);

- Annual production of each hydroelectric power plant for the years 2015, 2016 and 2018 (ODRÉ, 2015, 2016, 2018);

- Weekly hydraulic stock (Eq. (18)) at national level from 2014 to 2020 (RTE, c);

As mentioned in Appendix B, our final hydropower plants dataset does not include all the hydropower plants installed in France. However, using annual production data of each plant provided in (ODRÉ, 2015, 2016, 2018), we can quantify the share of the national production provided by the power plants in our database. This enables us to compute a factor to convert





the actual production of national time-series (RTE, a) into representative production in our model both for prescribing the production demand and comparing the results. The calculation of such conversion factors is presented in Table B2. It relies on the assumption that within each category of power plant, the geographical distribution of plants in our database is representative of all French power plants, so that production ratios remain constant over time. This assumption is debatable as our database includes the largest power plants in terms of installed capacity, which are predominantly concentrated in certain regions, while

smaller-scale plants may be located in watersheds not represented in our database (e.g., run-of-river plants on the River Seine for instance). However, as the missing plants have, by definition, a lower installed capacity than those in our database, their contribution to the national production is lower and can reasonably be neglected.

We compute the total energy storage capacity of the reservoirs associated with the power plants in our database using Eq. (18) and data from our plants and reservoirs databases. We obtain $S_{max} = 3.66$ *TWh*, which is quite close to the $3.59$ *TWh*

value reported by RTE (RTE, c). Therefore, we can consider that our database covers all the available storage and that missing hydropower capacity is linked to negligible reservoirs.

## 3.4 Hydro-meteorological errors

To evaluate the performance of the ORCHIDEE model to simulate river discharges in France, independent of reservoir operations, we compare daily river discharges simulated by the model with the observations database of Schapi (2022).

It is important to acknowledge that the observed discharge data represents actual discharge values, including water withdrawals, while at this stage, our model generates natural discharges without such withdrawals and dam operations.

### 3.4.1 Bias in average discharge

Figure 7 displays relative biases of average discharge simulated by ORCHIDEE forced by SAFRAN over the 2010-2020 period for a selection of gauging stations located on rivers equipped with hydropower infrastructure (see Fig. B2 for the detailed

locations of the power plants). We chose the bias metric because the annual mean discharge is the most relevant parameter is the most relevant parameter for hydropower potential.



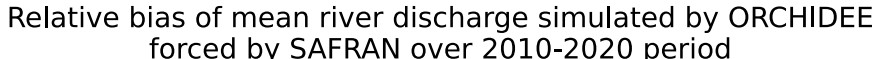

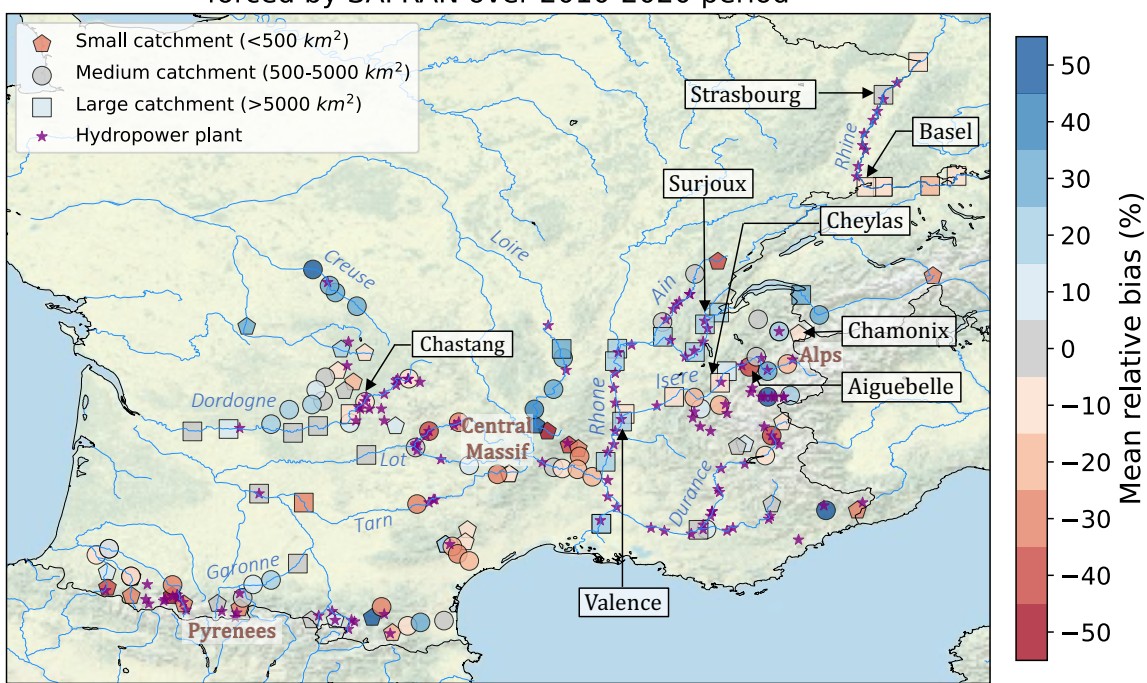

**Figure 7.** Relative bias of average discharge for a selection of gauging stations located on French rivers equipped for hydropower for period 2010-2020. Each colored point represents a gauging station, The shape indicates the size of the concerned watershed while the color indicates the calculated bias at this location. Purple stars indicate the locations of the hydropower plants located on the grid.

Source: authors, based on a layer by U.S. National Park Service

The overall performance of the model indicates a slight overestimation of flows, with an average bias of +2.4%.

The discharge bias shows an increasing trend with the upstream area of stations. For small catchments (less than 500 $km^2$),
the average bias is -1.6%. In medium-sized catchments (between 500 and 5000 $km^2$), the bias decreases to +1.1%. In large catchments (more than 5000 $km^2$), the bias becomes more pronounced, reaching 7.6%. It is however important to note that the smaller the upstream area, the greater the uncertainty in the location of the station. In Fig. 7, only the stations located with an error in the upstream area lower than 20% are displayed.

On the largest rivers (Rhine and Rhone), where most run-of-river power plants are located, the bias shows little spatial
variability, constant at around +20% for the Rhone and -10% for the Rhine respectively. In the Alps, on the other hand, where a significant proportion of dispatchable hydroelectric capacity is installed, the bias displays a high spatial heterogeneity, sometimes within the same river. Upstream of the Isere river, the bias varies from -19% to +26% between two stations some twenty kilometers apart. The upstream reaches of the Durance also show negative biases.





In the other massifs equipped for hydroelectricity (Pyrenees and Massif Central), there are also negative biases at altitude,
which gradually diminish downstream.

Assuming negligible observational errors, discharge bias can originate from different error sources:

– Errors in the atmospheric forcing applied to ORCHIDEE;

– Modeling errors in the energy, water and carbon cycles;

– Missing processes in ORCHIDEE like glacier melting, interactions with groundwater and water withdrawals).

To explore the first hypothesis, Fig. 8 compares discharges simulated by ORCHIDEE using the two alternative forcings
(SAF_COM and SAF_SPAZM) with the reference SAFRAN simulation. The relative bias of these simulations to observations
are presented in Fig. 9.

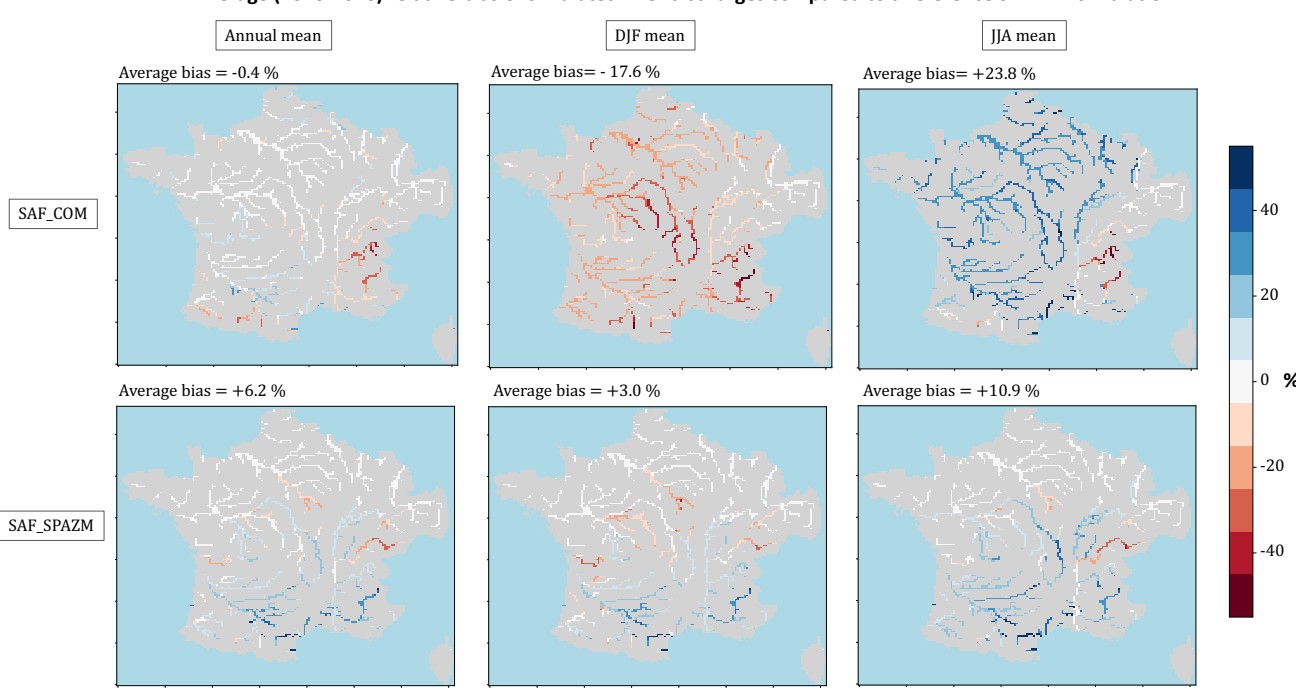

**Figure 8.** Average relative bias in discharge simulated by ORCHIDEE under alternative precipitation forcings. Results are given in relative
difference compared to the reference SAFRAN simulation, for the period 2010-2020. Left: annual average bias, middle: average bias in
Winter period (December-Januray-Febreuary), right: average in Summer period (June-July-August). The discharges are displayed for all grid
points with an upstream area higher than 1000 $km^2$.

Under SAF_COM, simulated discharges show relatively small differences on annual average, except in mountainous water-
sheds (Alps and Pyrenees), where the lower precipitation in COMEPHORE results in streamflows that are 30% to 40% lower





Relative bias of mean river discharge simulated by ORCHIDEE
forced by alternative forcings over 2010-2020 period

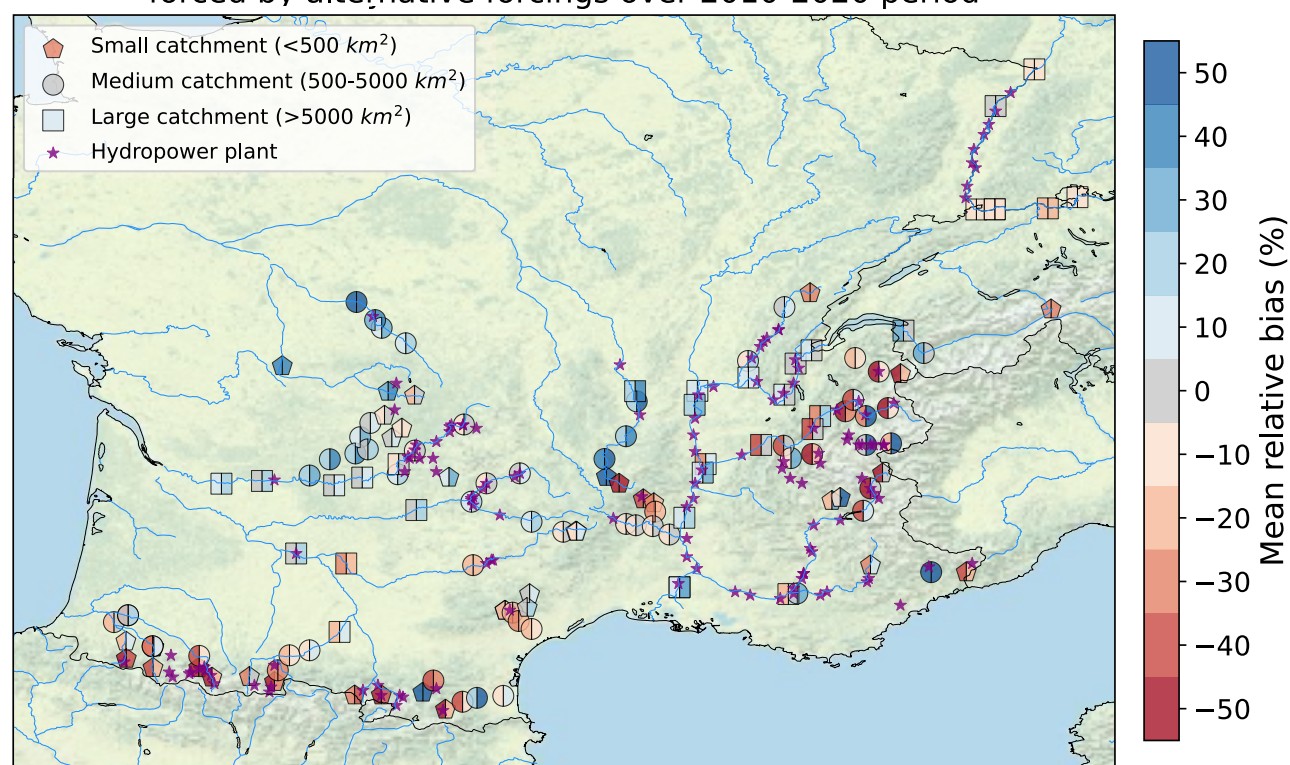

**Figure 9.** Relative bias of average discharge simulated by ORCHIDEE under alternative forcings for a selection of gauging stations located on French rivers equipped for hydropower for period 2010-2020. Left coloring indicated the average bias of discharges simulated under SAF_COM while right coloring indicated the average bias of simulations under SAF_SPAZM.

Source: authors, based on a layer by U.S. National Park Service

when compared to the SAFRAN simulation. However, a pronounced seasonal pattern is observed. The simulated streamflows in winter are lower in the simulation forced by COMEPHORE across France (averaging -16% and up to -50% for the Loire and Durance rivers), while in summer, they are higher (averaging +25% and up to +50% for the Loire River). As regards comparison with observed flows, the negative biases existing under SAFRAN in the Alps and Pyrenees are accentuated, particularly along the Durance and Isere rivers where many hydroelectric power plants are located. However, for some Alpine

stations and the Massif Central, for which the flow is overestimated with SAFRAN, the flow is more accurately simulated with COMEPHORE.

Under the SAF_SPAZM forcing, river discharges show an increase in the majority of watersheds, which is consistent with the previously highlighted higher precipitation in this dataset. However, the upper Rhone watershed stands out with a decrease





in simulated discharge, reaching up to -40% during the summer season, allowing for a reduction in the bias of simulated
discharges in this area.

Even if we limit our analysis to the precipitation variable without considering other forcing variables, we show a significant
influence of the forcing variability on the simulated discharges.

### 3.4.2   Discharge seasonality

Beyond the bias in average values, the performance of ORCHIDEE in reproducing the seasonality of the discharge is key for the
modeling of run-of-river production as well as that of poundage power plants, which have only a very limited storage capacity.
Observations and simulations of daily discharges under SAFRAN forcing are presented in Fig. 10 for selected gauging stations
in catchments equipped with run-of-river or poundage power plants.

As depicted in Fig. B2, run-of-river plants are mostly located along the Rhone and Rhine rivers. In the upper-Rhone (Surjoux
station), there is a substantial overestimation of high flows and underestimation of low flows. The error reduces progressively
downstream: the Nash Sutcliffe efficiency (NSE) is better at Valence station, despite a higher overall annual bias (likely due
to the non-representation of water withdrawals). On the Rhine (Basel and Strasbourg stations), we see similar errors, with an
underestimation of low flows during the Fall and an underestimation of the Spring maximum. The discrepancy in the Rhone's
seasonality can be attributed to the non-representation of Leman reservoir management in our model, which is known to play
a crucial role in shaping discharge seasonality in the upper Rhone (Habets et al., 1999).
Poundage plants are distributed across various catchments. Some of them are concentrated in the upper Dordogne river,
notably the Chastang plant, the most powerful poundage facility, which benefits from a gauging station at its location. We find
a positive NSE for this station, indicating that the seasonality is well captured by the model.

Finally, some run-of-river and poundage plants are also concentrated in the Alps, where we focus on two gauging stations:
Chamonix, situated in a small upper catchment, close to a run-of-river plant and Cheylas, positioned on a large river (l'Isère),
downstream from several power plants. At Chamonix, we find a seasonal bias as the model simulates an earlier discharge
peak compared to observations (around 2 months ahead). At Cheylas, the model overestimates the seasonal variability of the
discharge, with higher flows during Spring and lower flows during Winter, which can be attributed - at least in part - to the
non-representation of reservoir management at this stage of our study (see Sect. 5.3).

### 4   Calibration of the hydropower operation model

The hydro-meteorological biases highlighted in the previous section can lead to significative errors in the estimation of hy-
droelectric production. However, the limited knowledge of actual hydropower networks (unknown values for plant efficiency,
uncertainty of the water input of reservoir plants) can also contribute to similar errors.

In this section, we estimate the differences between AHPs (Eq. (19)) simulated by the model, and the observed annual
production at each power plant (as mentioned earlier, this data is available for the years 2015, 2016, and 2018). We discuss the





**Figure 10.** Comparison of simulated and observed river discharges for a selection of gauging stations. Locations of selected stations are indicated in Figure 7. Fines lines and dots are daily time-series while ticker lines are 30-days sliding average. NSE metrics are computed on daily time-series.

likely origin of these differences and propose a calibration method for the unknown parameters to address these differences, to make the hydrological cycle simulated in ORCHIDEE consistent with observed production.

We first describe the calibration approach for run-of-river power plants, and then for reservoir power plants. Finally, we validate this calibration by comparing annual potentials simulated in ORCHIDEE to observed annual production at the national level on an extended period (data available from 2000 to 2020).

We choose to use SAFRAN forcing as a reference for the calibration step, as this dataset is widely used in regional studies of France.





## 4.1 Run-of-river plants

As curtailment of run-of-river production is generally not assumed, AHPs (Eq. (19)) should therefore be a good approximation of the observed production. A bias in simulated AHP of a run-of-river plant compared to its historic production can be explained by five reasons:

1. Hydro-meteorological bias leading to different river discharges in the model. These errors have been assessed in the previous section;

2. An inexact location of the hydropower plants during the placement on the HTU graph, which leads to over/under estimation of the available discharge at the plant location. However, Fig. 6 shows that the error is less than 10% for most of the plants we placed on the river network;

3. We assume that all the water of the river can be exploited by the plant. In reality, the river can be divided into several branches with only one of them passing through the plant;

4. Plants efficiencies are assumed to be equal for all plants and constant to 0.9. In reality, the efficiency of a hydropower plant depends on the type of hydroelectric turbine that is used (the choice is made based on the plant's rated head and flow) and varies with the flow rate;

5. We assume that plants produce at their maximum potential. However, in reality, a plant can be unavailable for some period - due to maintenance, for instance. Moreover, some of the plant's potential can be reserved for ancillary services to the grid. This can reduce the actual production compared to the potential.

Fig. 11 shows the average relative bias in simulated AHP compared to observed production for the three years with available data for the run-of-river plants in our database. For most of the plants, bias in hydropower potential is close to the bias in discharge computed at neighboring stations displayed in Fig. 7, meaning that it mainly comes from the hydro-meteorological error (reason 1). At Caderousse (on Rhone) and Gambsheim (on Rhine), a stronger positive bias is found. At these locations only part of the river is passing through the plant (reason 2).

As in previous studies (Wagner et al., 2017; Zhou et al., 2018), the unknown efficiency of the power plant $\eta_{(i,j)}$ is adjusted to calibrate the model to the historical annual generation data based on previously estimated bias (Eq. (20)). Such calibration corrects the total error without differentiating its source.

$$\eta_{(i,j)} = \frac{1}{0.9} * \overline{\frac{E_{(i,j)}(y)}{AHP_{(i,j)}(y)}} \tag{20}$$

Obtained efficiencies range from 0.43 to 1.31 with a median value of 0.88.



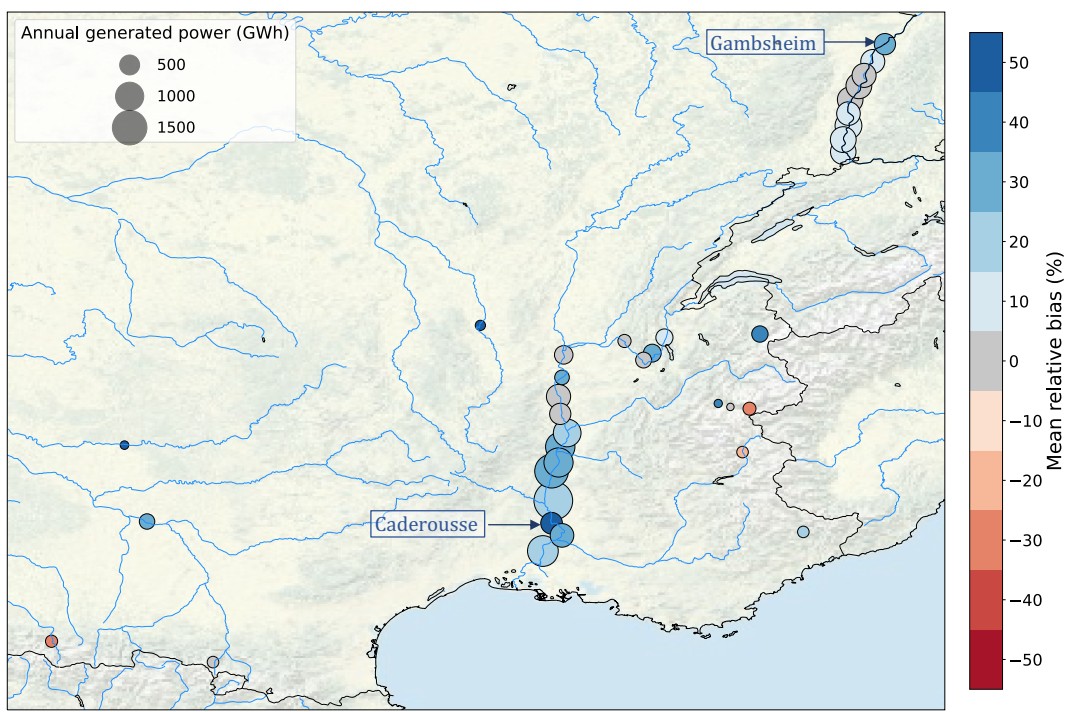

**Figure 11.** Average relative bias (in %) of simulated annual hydropower potential compared to observed historic production for run-of-river plants with available data. Point size corresponds to average annual production.

Source: authors, based on a layer by U.S. National Park Service

## 4.2 Reservoir plants

Over a year, all the water entering the reservoir of a plant could indeed either contribute to the annual production of the plant $E_{(i,j)}(y)$, to the annual change of the hydraulic stock in the reservoir $\Delta S_i(y)$ or spill without generating power.

Observed production $E_{(i,j)}(y)$ is available for the three years mentioned earlier, however observations of the change of the hydraulic stock are only available at the national level for the national stock $\Delta S_{obs}(y) = \sum_{i \text{ in res}} \Delta S_i(y)$. To compare simulated AHPs with observations of production and stored energy, we make the two following assumptions: (i) spillages that do not produce power can be neglected and (ii) the change in hydraulic stock is homogeneous across all reservoirs: $\forall i, \Delta S_i(y) = \Delta S_{obs}(y) \times \frac{S_{max}}{S_{i,max}}$.

In Fig. 12, we plot the average bias of $AHP_{(i,j)}(y)$ relative to observed net production $E_{(i,j)}(y) + \Delta S_i(y)$ for the three years for which data is available. It enables us to distinguish two types of bias in the simulated AHP, suggesting that two main error sources can be distinguished:





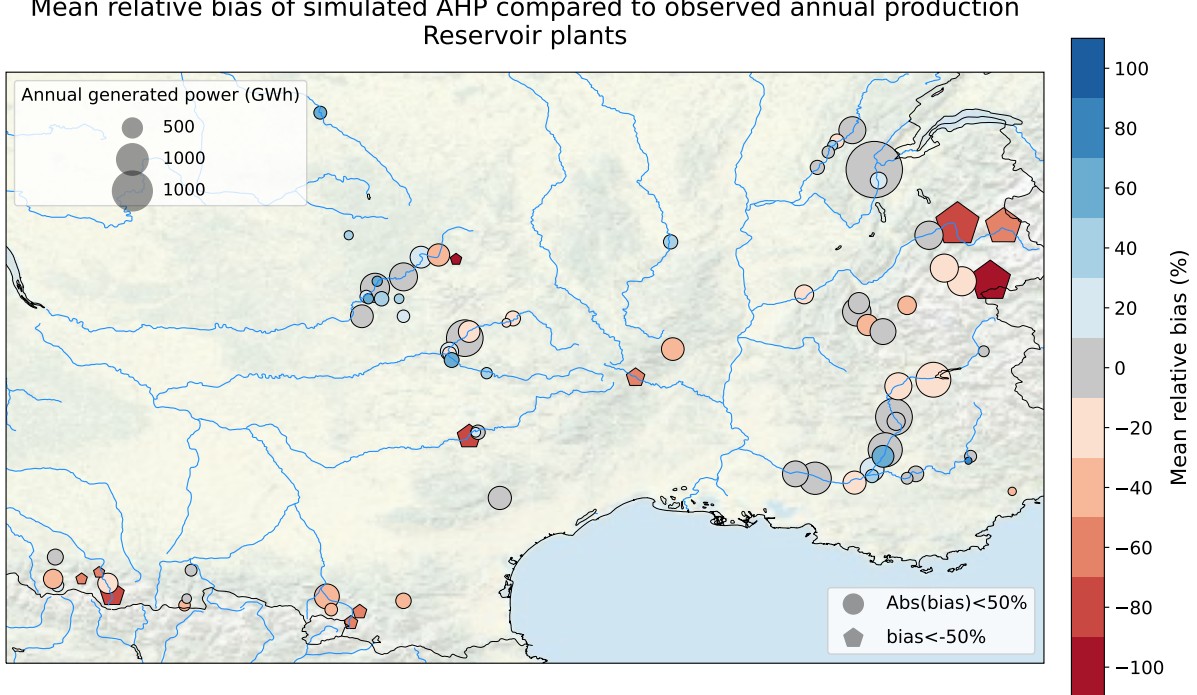

**Figure 12.** Average bias in simulated AHP compared to observed historic net production of reservoir plants with available data. Points size correspond to the average annual production of the plant.

Source: authors, based on a layer by U.S. National Park Service

- Plants that have an absolute bias inferior to 50% (represented by circles in Fig. 12). Their biases are generally similar to the one of discharge for neighboring stations in Fig. 7.

- Plants that have an absolute bias superior to +/- 50% (represented by pentagons in Fig. 12). These plants are mainly located in mountain areas and have a negative bias larger than the the one of the discharge in this area. Moreover, their biases have a small interannual variance, indicating that the error is stable in time.

As for run-of-river plants, differences in simulated AHP compared to observed production of reservoir plants can have different sources. In addition to the five errors listed above that apply also to reservoir plants, a sixth possible error, related to the adduction network, should also be considered. Indeed, we assume in our model that each plant is only fed by one reservoir, which can lead to an under-estimation of the plant production if some other water inputs are non-negligible.

For the first category of plants, the first reason (hydro-meteorological error) can explain most of the computed bias. For the second category however, the sixth reason (error on the adduction network) seems to be predominant. Indeed, mountain plants are fed by several water intakes, and thus the intake reservoir we are considering, even if it represents the main source of water, is only a small portion of the water input. Therefore, the simulated inflow at the reservoir location is not sufficient to generate





the observed production, which can explain the large negative bias we find. The example of La Bathie power plant in the Alps
is detailed in Appendix D.

To account for these two different error sources, we calibrate the model in two successive steps:

- Step 1: Dams with a large negative bias (inferior to -50 %) are shifted from their original location to take into account the
computed deviation (they are located as if their upstream area was corrected by the ratio). This allows for the modeling
of an entry point of the power plant which receives enough water. Most concerned areas are located in mountains,
where the water intakes are quite close geographically (on the same atmospheric grid) and therefore subject to the same
precipitation, which legitimizes this operation.

- Step 2: Once the network error is corrected, the efficiencies of the plants are adjusted to match the observed production,
as is with run-of-river. We find a median efficiency of 1.01.

### 4.3   Validation of the calibration

The performance of the calibrated model is assessed by comparing potentials simulated by the calibrated model forced by
SAFRAN with the historic annual production (RTE, a) for the different categories of power plants over the whole period
(2010-2020) (Fig. 13). We assume that hydropower is used as much as possible and that the production is well managed, so
that actual production is a good proxy to compare with our potential production AHP. To align with RTE's classification, we
categorize the reservoir power plants into two groups: poundage and reservoirs.

For a given category, the annual simulated potential is computed by summing AHPs of all plants belonging to this category. For
poundage and reservoir plants, we directly compare this aggregated potential to the historic production, as stock data (RTE, c)
is not available for the whole period. This relies on the assumption that the national stock returns to its initial value at the end
of each year.

The calibration appears to be robust as a very small bias (less than 3%) is obtained when comparing the simulated potentials to
the observed production. The relative differences of annual production are on average lower than 10%. This indicates that the
model is able to capture the overall pattern of interannual variability of the observed production.

We also explore the sensitivity of our model and calibration procedure to the uncertainties in precipitation forcings high-
lighted in Sect. 3.4. We compute annual hydropower potentials under the two alternative forcings and compare the inter-annual
variability of observed production to the inter-forcing variability (Fig. 13).

Run-of-river annual potentials exhibit little variation across the different forcings, as the simulated flows of major rivers
hosting run-of-river power plants (primarily the Rhone and the Rhine) demonstrate a low sensitivity to atmospheric forcing.
Consequently, the inter-forcing variability of simulated potential (defined as the mean standard deviation of annual potential
across the forcings) is three times smaller than the interannual variability of run-of-river power production(defined as the
standard deviation of observed annual productions), see Table 3. It is also slightly smaller than the modeling error (RMSE of





SAFRAN simulated potentials compared to observations), indicating a low sensitivity of simulated run-of-river production to the atmospheric forcing.

Conversely, reservoir plants production shows a much higher sensitivity to precipitation disparities between forcings. Lower COMEPHORE precipitations in mountainous regions lead to an average decrease of 18.7% in the total simulated potential, 605 compared to the SAFRAN simulation. As a result, the variability among forcings is of the same order of magnitude as the interannual variability of production and higher than the modeling error.

Finally, poundage power plants fall in an intermediate category, displaying an inter-forcing variability that is 41% lower than the interannual variability.

In conclusion, the uncertainties in precipitation forcing in mountainous regions prove to be critical in the estimation of 610 realistic hydropower potentials for reservoir plants. The calibration carried out relative to SAFRAN is less effective for other forcings (SAF_COM for instance), as the differences in precipitation data appear as the main contributor to the differences in hydropower potentials.

**Simulated annual hydropower potential compared to observed production**

![Figure 13: Line and scatter plot titled "Simulated annual hydropower potential compared to observed production." The y-axis shows Production (TWh) from 5 to 45, the x-axis shows years from 2000 to 2020. Legend indicates Observations (black), SAFRAN (blue), SAF_COM (green), SAF_SPAZM (orange), with markers for Run-of-river (circle), Poundage (diamond), and Reservoir (triangle).]

**Figure 13.** Comparison of estimated annual hydropower potential with observed annual production for the different categories of hydropower plants and for the different atmospheric forcings, after calibration based on SAFRAN.

## 5 Validation of the hydropower simulations

In this section, we assess the model's ability to simulate reservoir management and hydropower production. Observed time-615 series for each type of hydropower plant (run-of-river production, reservoir production, PHS production and pumping) serve as demand inputs for the reservoir operations in the model. At each time step, the model aims to meet this target by operating





| | Run-of-river | | Poundage | | Reservoir | |
|---|---|---|---|---|---|---|
| | Calibration Period | Validation Period | Calibration Period | Validation Period | Calibration Period | Validation Period |
| Mean relative error | - | + 2.8 % | - | -2.6 % | - | -1.4 % |
| Mean absolute relative error | 3.5 % | 6.9 % | 3.7 % | 5.4 % | 2.5 % | 7.5 % |
| Interannual variability (TWh) | 3.71 | | 1.71 | | 2.61 | |
| Inter-forcing variability (TWh) | 1.32 | | 1.25 | | 2.54 | |
| Modeling error (TWh) | 2.64 | | 0.67 | | 1.33 | |

**Table 3.** Estimation of the errors in annual potentials prediction

the reservoirs according to the rules described in Sect. 2.3 and the simulated hydrological cycle. The objective is to verify if our model is able to simulate operations consistent with observed production and pumping.

We present here the results obtained from a simulation spanning the period from 2015 to 2020 (period of availability of validation data).

## 5.1 Run-of-river production

30-minutes run-of-river production observations (RTE, a) include production data from real run-of-river plants that do not have reservoirs, as well as poundage plants associated with small reservoirs. We reproduce this distinction in our model by gathering these two kinds of plants.

At each time step, the model first computes the available potential from fatal production (from run-of-river plants and spill or constrained releases from the reservoirs of poundage plants). If this potential falls short of fulfilling the production target, it then operates the reservoirs associated with poundage plants to supplement the production.

Figure 14 details how the simulation compares to the prescribed production throughout the period when forced by SAFRAN. The overall seasonality of the production is quite well reproduced, with the model succeeding in meeting the hourly production target 69.0% of the time. The failures represent a total volume of 6.9% of the prescribed production over the six years. These failures (in red in Fig.14) mostly occur during Summer and Fall and indicate that the simulated hydrology is unable to produce what was actually produced during these periods. In Winter and Spring, however, there are instances when the potential of fatal production is higher than the target production (January and February 2018 for instance), which means that, in the model, more power could have been generated during these periods than was actually observed. These discrepancies are likely due to the discharge seasonality bias in the Rhone and Rhine catchments highlighted in Sect. 3.4.2.

Despite these discrepancies, the performance of the model remains satisfactory, as it captures gross seasonality and magnitude of run-of-river production, in addition to the inter-annual variability (Fig.13).

Simulation of run-of-river production in the model, when forced by the alternative forcings SAF_SPAZM and SAF_COM, are presented in Fig. C2 and C4. Using SAF_SPAZM, the failures in meeting the prescribed production are reduced (4.3% of



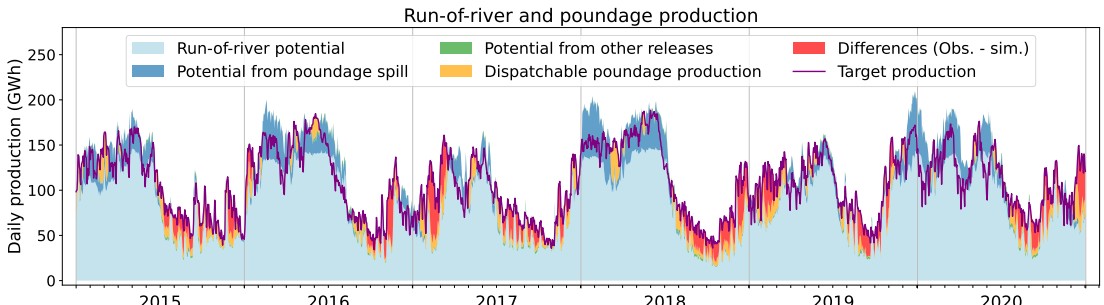

**Figure 14.** Run-of-river and poundage plants daily production. Purple line indicates the production prescribed to the model and red shows the difference between this production and the one simulated in the model when forced by SAFRAN. The other colors refer to the nature of the flow that contributes to production in the model. Light blue represents the gross potential of run-of-river plants, dark blue represents the potential of spill from poundage reservoir (water overflowing from the reservoir), green represents the potential from constrained releases of poundage reservoirs and lastly orange represents the dispatchable production, generated by the water specifically released from the poundage reservoirs for power generation.

production not satisfied compared to 6.9%), due to slightly higher annual potentials of run-of-river and poundage power plants (Fig. 13). On the other hand, with SAF_COM the lower potentials lead to higher failures (15.4% of the total production), consistent with the lower potentials obtained in Fig. 13. However, the seasonality remains very similar in all three simulations, consistent with the similar seasonality of the simulated discharges for the Rhine and Rhone rivers (Fig. 8).

## 5.2    Reservoir production

Similarly, 30-minutes time-series of observed production from reservoir power plants is prescribed to the model. To fulfill this demand, the model completes the fatal production that may be available from reservoirs spillage and constrained releases by operating reservoirs according to rules defined in Sect. 2.3.

    The daily reservoir hydropower production simulated under SAFRAN is compared to the historic production in Fig. 15 over the entire simulation period. Simulated production under the other forcings are presented in Fig. C3 and C5. Figure 16 displays
the co-evolution of the observed national hydraulic stock (RTE, c) and the one simulated in the model (Eq. (18)) for the three forcings under study.

    Under SAFRAN, the model successfully meets the production target while simulating hydraulic stock variations consistent with observations throughout the 6 years. In the model, reservoirs are indeed filled during Spring due to snow melt and depleted during winter to meet the high electricity demand. Annual variations of hydraulic stock are in line with the discrepancies high-
lighted in Fig. 13. For example, Fig. 13 shows that the simulated annual potential assuming stock equilibrium is slightly higher than the observed production for the year 2016. Consequently, as we constrain our model to match the observed production, it results in a small volume of water being stored during this year, leading to a stock higher at the end of the year than at its beginning, which is what we actually observe in Fig. 16. On the contrary, the annual potential simulated assuming stock



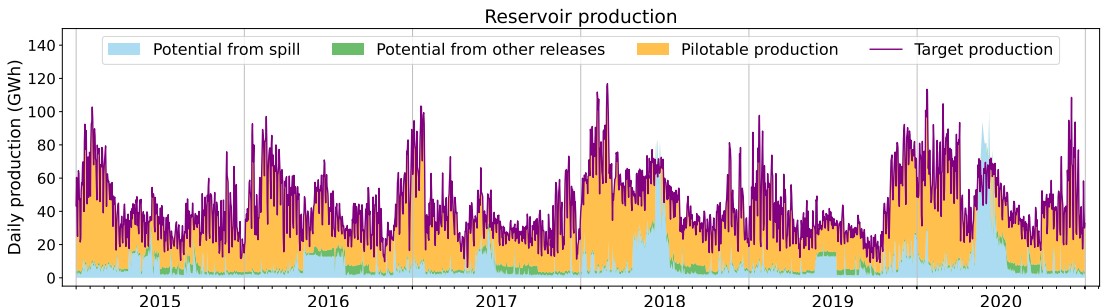

**Figure 15.** National reservoir plant production simulated in the model. Purple line indicates the production prescribed to the model, while the other colors refer to the nature of the flow that contribute to this production. Blue represents the gross potential from reservoir spillage (water overflowing from the reservoir), green represents the potential from constrained releases of the reservoirs and lastly orange represents the production by the water that is specifically released from the reservoir for hydropower purpose.

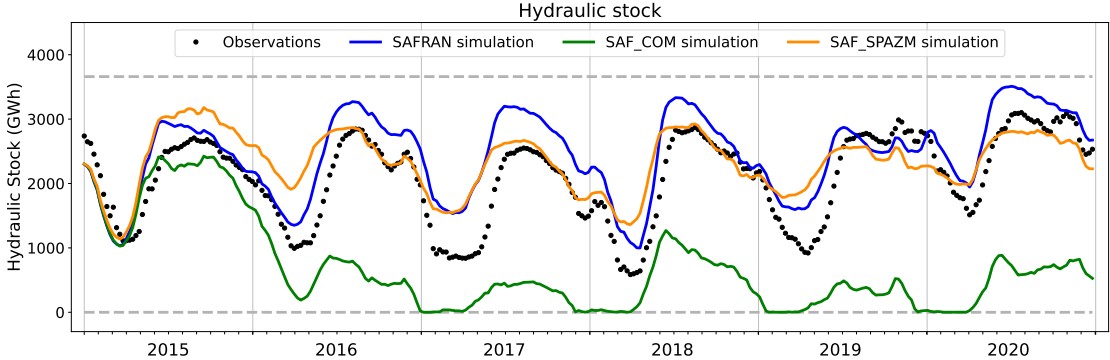

**Figure 16.** Comparison of national hydraulic stock evolution simulated by the model and weekly observations

equilibrium in 2017 is lower than the observed production in Fig. 13, explaining why we simulate a negative annual stock
variation over this year (Fig. 16).

Nevertheless, a slight temporal shift is observed, as the simulated stock starts to fill some weeks earlier compared to the observations. This shift aligns with the seasonal biases in river discharges identified at the Chamonix Station (Fig. 10), indicating a consistent pattern.

Under SAF_SPAZM, the evolution of the simulated stock remains quite satisfactory in comparison to observations, demonstrating a certain robustness of our model relative to changes in precipitation forcings. As shown in Fig. C3, the production potential from reservoir spillage exceeds the target production on multiple occasions (e.g. during May and June 2016). This leads to unused water releases from the reservoir, explaining the negative annual variation of hydraulic stock we obtained for





2016, despite the simulated annual potential being greater than the observed production in 2016 (Fig. 13).
Under SAF_COM, however, the stock is completely emptied after the two first years of simulation, and a significant portion of
the demand cannot be satisfied (Fig. C5), consistent with the huge difference in annual production estimates highlighted in Fig.
13. In addition to the substantial deficit in hydropower potential, a negative feedback loop comes into play. As the reservoir
storage diminishes, the head of the power plants decreases, consequently reducing the associated power generation for a given
released volume. Consequently, the power plants draw more water to generate the same amount of energy, further exacerbating
the decline in reservoir storage. The calibration carried out relative to SAFRAN is not effective to avoid this outcome.

Figure 15 allows for the distinction of the different drivers of French hydropower production, depending on the season. In
winter, hydropower production is substantial, driven primarily by high electricity consumption. The majority of production
stems from intentional reservoir operations, with a minimal proportion attributed to fatal production. In spring, fatal produc-
tion becomes more prominent, particularly due to snow melt-induced spillage, resulting in a minimum hourly production, even
during periods of low consumption such as at night (only visible at the hourly resolution not displayed here). During summer,
although there is no spillage, a significant portion of the hydropower potential comes from constrained ecological and agricul-
tural water releases. When looking at the hourly production (not displayed here), we find a good agreement of the simulated
minimal production with the observed troughs in RTE's production.

**5.3   Effects of hydropower operations on river discharges**

In Sect. 3.4 we identified biases in river discharge seasonality, in particular for stations in the Alps (Cheylas station, Fig.
10), which could possibly be attributed to the non-representation of water management. We explore in this section how the
representation of hydropower operations can reduce these biases, with the example of two gauging stations located in the Alps.
Figure 17 details the location of these stations comparatively to hydropower infrastructures and adduction network. Aiguebelle
station is located on the Arc river, just upstream of its confluence with the Isère, and downstream from a series of hydropower
plants, including one that generates electricity thanks to the release of a dam on the Isère river. Cheylas station is located on
the Isère river, downstream of its confluence with Arc.

Figure 18 compares the seasonality of the discharges simulated at these two locations by ORCHIDEE forced by SAFRAN
with and without activating the hydropower operations module.

At Aiguebelle, the representation of this inter-basin water transfer significantly reduces the annual bias from -31% to -4%
(Fig. 18). Indeed, when hydropower operations are activated, part of the Isère's water is diverted from its natural outlet to
supply a power plant on the Arc. At Cheylas, no change is observed in the bias of the simulated river discharge.

Discharge seasonality is improved for both stations with higher flows in Fall and Winter due to releases for power generation.
The NSE metric is therefore significantly improved.

We found a similar effect for other French watersheds where flow observations near hydropower plants are available. How-
ever, as mentioned earlier, the professional secrecy surrounding French hydroelectric production complicates a systematic and
precise evaluation of this improvement in flow simulation.





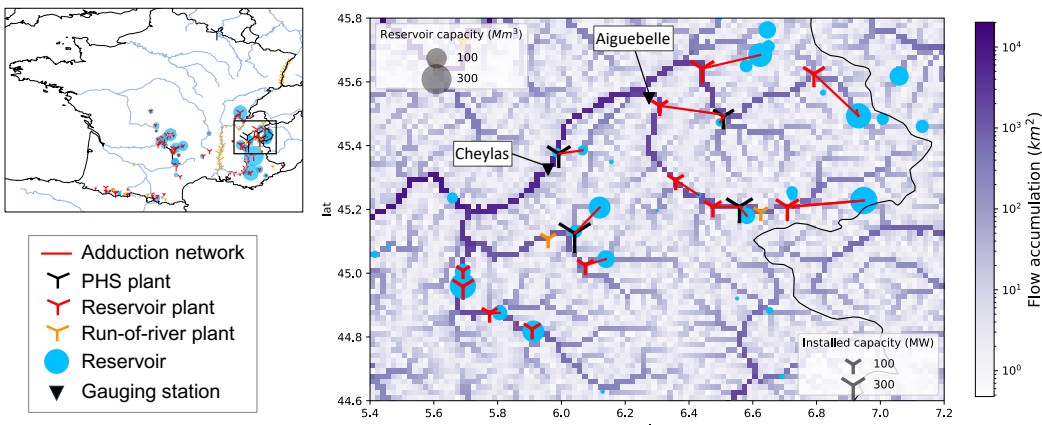

**Figure 17.** Location of Aiguebelle and Cheylas stations comparatively to hydropower infrastructures in Arc catchment (French Alps)

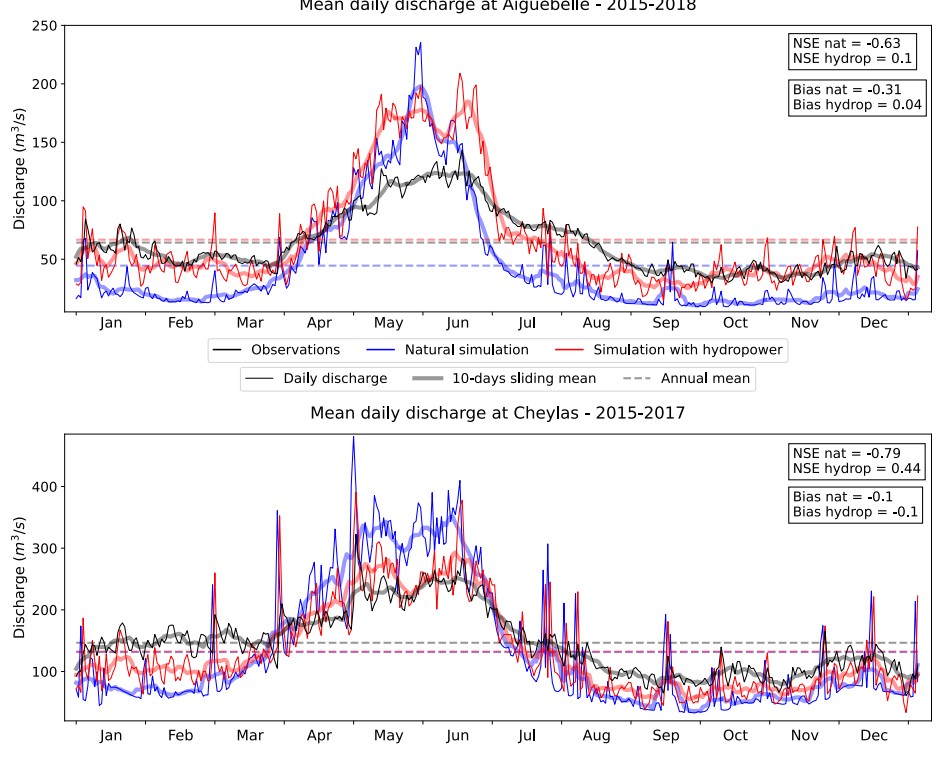

**Figure 18.** Comparison of daily (fine line) simulated river discharge with hydropower operations (red) and without (blue) and observed discharge (black) for two gauging station in the French Alps. Thicker line is the 10-days average while dashed line is the annual mean.



# 6    Discussion and conclusion

## 6.1    A demand-based approach

In this study, we demonstrate the effectiveness of a demand-based approach to simulate hydropower operations in land surface models.

The conceptual framework of such an approach is first described, emphasizing its three original features: (i) the reconstruction of the human-made hydropower network on the model grid to represent not only natural water flows, but also those build for hydropower management; (ii) the implementation of reservoir operation rules that account for their multi-purpose objectives;

(iii) the prescription of an exogenous "hydropower demand" defined at the power grid level to drive the release rules of hydroelectric reservoirs, allowing coordinated management of all hydroelectric resources on the power grid and consistent with power system needs.

Then, we explore the performance of this approach when implemented in the routing module of the ORCHIDEE model, for the case study of the French power grid. ORCHIDEE is run forced by an atmospheric reanalysis dataset and national historic

hydropower production time-series are prescribed to the model as the hydropower demand to satisfy. We find that, when forced to reproduce the historic generation, the implemented method simulates hydroelectric reservoir operations in line with observations of reservoir storage at the national level.

Beyond this satisfactory result, our method presents several limitations and opportunities for improvement.

First, the time-series used to drive the reservoirs releases in this study is the actual production of dispatchable hydropower plants, which can differ from the real demand for dispatchable hydropower production. Indeed, the actual production results from a trade-off between the demand and the prevailing hydrological conditions, particularly the current storage level in reservoirs. If this storage is low, the demand would not be completely satisfied in order to maintain a certain level for future uses.

Besides, we consider an exogenous dispatch of the hydropower production across the different types of hydropower plants

(namely run-of-river and reservoir) at each time step. This allows us to more easily identify model weaknesses for each type of power plant. For instance, we found a seasonal bias in run-of-the-river hydropower production, that we would have missed if a single production target had been used for all power plants. Reservoir plants would have acted as buffers, reducing their production during excess run-of-the-river output and increasing it during deficits, so that we would have observed discrepancies in the stock evolution. However, in reality, the dispatch of power demand across the different types of hydropower plants is not

exogenous but also depends on the hydrological conditions, the potential for run-of-the-river production being fully exploited before turning to dispatchable units.

To capture these intricate interactions between hydrology and hydropower production decisions, a solution is to couple our model with an economic power system dispatch model (Oikonomou et al., 2022). This coupling would ensure that the power demand dispatch used to drive reservoir operations in ORCHIDEE considers the hydrological states simulated within the

ORCHIDEE model. It would result in a comprehensive modeling framework wherein simulated hydropower production simultaneously adheres to constraints related to water availability, non-power reservoir operations, and minimization of power





system costs. In particular, hydropower demand would be endogenously adjusted to match the hydropower potentials of the simulated hydrology, and could avoid entering the feedback loop where reservoirs are emptied, as in the SAF_COM simulation. This novel approach holds significant promise for enhancing the consistency and realism of hydropower production simulation, in particular the study of the joint impacts of climate change and variable renewable energy integration.

Second, in this study we opted for a simple rule to distribute national production among different power plants and demonstrated that such a rule can simulate credible hydroelectric operations at the national level. As no time-series of production is available at the individual plant level in France, the realism of the simulated individual operations is difficult to assess. This choice can, however, be further investigated, in particular by testing alternative distribution rules, such as those proposed by Lund and Guzman (1999).

Additionally, the operations we simulate assume that a social planner controls the entire grid's power plants and reservoirs, optimizing the collective production. In reality, power plants may belong to different stakeholders, each seeking to maximize their profit. Ambec and Doucet (2003) have shown that such decentralized management can lead to suboptimal resource management, which could not be reproduced by the proposed model. However, in the case of France, our assumption is justified as the historical production company, EDF, owns nearly 85% of the hydroelectric production.

Third, as we focused primarily on hydroelectric usage, other water uses are simplified or even absent in the current version of our model. Specifically, no water abstraction for domestic, industrial, or agronomic needs is included in our model. Following Zhou et al. (2021), the irrigation demand could be explicitly calculated by the model based on the deficit between potential evaporation and actual evapotranspiration. In other studies, domestic and industrial water demands are estimated using socio-economic proxies such as population density or GDP (Neverre, 2015).

As mentioned in the introduction, process-based hydrologic models forced by future climate projections have been used to project the evolution of annual hydropower production at a national or regional scale (Van Vliet et al., 2013; Liu et al., 2016; Voisin et al., 2020; Chowdhury et al., 2021). By differentiating the place of water abstraction from the place of power generation, our new method offers opportunities to improve the estimation of such annual potentials. To accurately assess the resilience of power grids to a different climate, a realistic representation of daily and hourly hydropower operation is required. In most previous hydropower studies, hourly hydropower production is calculated based on reservoir releases that follow generic rules, independently of the specificity of hydropower reservoirs (Abeshu et al., 2023). Our method instead, allows hydroelectric dams to be specifically operated, in line with hydroelectric demand, and thus improving the representation of the hourly production of dispatchable hydropower plants.

## 6.2 Sources of uncertainties

We have paid particular attention to identify and discriminate among the various sources of uncertainty that may affect the estimation of hydroelectric production using such a method. We demonstrate that, while errors in simulated discharge are predominant for most watersheds in our case study, our limited knowledge of the hydroelectric adduction network is the main source of uncertainty for hydropower infrastructures in mountainous basins. To our knowledge, no dataset comprehensively





documents these complex "hydroelectric links", which operate on a small scale. Therefore, an in-depth analysis of the gray literature released by the various stakeholders is necessary to reconstruct this network in detail. We nevertheless propose a calibration method to overcome this limitation, and validate it against observations for the case study of France. This method can therefore be extended to countries with little information available on the hydroelectric network.


Regarding hydro-meteorological errors, the use of three different precipitation datasets allows us to understand their more precise origin. In several watersheds crucial for hydroelectricity (such as Durance or Lot), and especially in the upstream parts, uncertainties in observed precipitation appear to be the primary contributor to the error in simulated discharge. On the Rhone or the Rhine rivers, on the contrary, errors in the simulated discharges seem to stem more from processes not represented in the

model (such as water withdrawals for human uses, for example).

Though incomplete, this work contributes to the current effort to integrate human water management into hydrological models, in order to simulate a more realistic water cycle (Nazemi and Wheater, 2015a). We show that our method can improve river flow simulations in some mountain catchments where hydropower cannot be neglected.

Finally, our study shows that comparing hydropower estimates with observed production offers an indirect means of checking the quality of meteorologic data. In our study case, we demonstrate the lower quality of COMEPHORE dataset in mountainous regions compared to SAFRAN or SPAZM, something already identified by Birman et al. (2017); Magand et al. (2018).

## 6.3 Perspectives

In conclusion, the demand-based operations proposed in this study hold promising prospects for enhancing our understanding

of the resilience of different power mix scenarios to changes in climate, water management or land use. The next steps in this trajectory involve (i) integrating our climate-based hydropower model with a power system model to get a comprehensive framework that captures all relevant constraints on hydropower production, (ii) applying this integrated framework to climate change scenarios and power system scenarios to assess the adaptive capacity of the power grids, and (iii) refining the description of other water uses to more completely describe the competition for water resources.

*Code and data availability.* The ORCHIDEE version developed for this project is available upon request. The meteorological forcings used in this study were provided by Meteo-France for SAFRAN (https://www.umr-cnrm.fr/spip.php?article788&lang=en) and COMEPHORE (https://radarsmf.aeris-data.fr/en/home-page/), and EDF-DTG for SPAZM (Gottardi et al., 2008)). The observed data used for validation is openly accessible online. River discharge data can be downloaded at https://hydro.eaufrance.fr/, while data on energy production is available at https://opendata.reseaux-energies.fr/. The reservoir dataset was built based on the GRanD database (Lehner et al., 2011), which can be

found at https://www.globaldamwatch.org/grand/, and on the data of the *Comité Français des Barrages et Réservoirs* (CFBR) at https://www.barrages-cfbr.eu/-En-France-. Finally, the plants database was built from the EU JRC hydro-power plants database (https://github.





com/energy-modelling-toolkit/hydro-power-database) and the *Registre national des installations de production raccordées au réeau de transport d'électricité*, which can downloaded at https://opendata.reseaux-energies.fr/.



## Appendix A: Locating hydroelectric infrastructures on the river network

Dams and hydropower plants are located on the MERIT grid based on geo-referenced and upstream area information provided in the databases (Infrastructures datasets used for our study over France are presented in Appendix B). The location procedure is done following these steps:

1. We identify a first location based on the infrastructure's geographical coordinates.

2. We define a search area around this first location (typically 10km)

– If the upstream area of the infrastructure is informed in the databases, we identify all the pixels in the search area having an upstream area close enough to the one being searched (typically +/- 20%) and, among these eligible pixels, the one closest to the first location is selected. If no pixel checks this condition, the infrastructure is not placed.

     – Otherwise, we look for the closest pixel to the first location likely to be positioned on a river. To do this, the
maximum upstream area of the pixels in the search area is identified ($U_{max}$) and the closest pixel to the first guess pixel satisfying ($U > \frac{U_{max}}{10}$) is selected, with $U$ being the upstream area of the pixel.



## Appendix B: Datasets

### B1 Dams and reservoirs

We use global reservoir data from GRanD (Global Reservoirs and Dams) dataset (Lehner et al., 2011), that gathers data of large
reservoirs and dams worldwide (volume $> 0.1 km^3$, hence a total of 7320 dams). The database contains 137 dams in France, 63
of which are used for hydroelectricity. However, some important dams for French hydroelectricity are not documented in this
database. Therefore we completed the database for this study with data from the CFBR (Comité Français des Barrages et des
Réservoirs), which is in charge of the inventory of French dams higher than $15m$ for the ICOLD (International Commission on
Large Dams). We extracted data from its website https://www.barrages-cfbr.eu/ to complete the GRanD database. Our database
finally gathers 492 French dams (Metropolitan France). Their location, original database and intended purposes are shown in
Fig. B1.

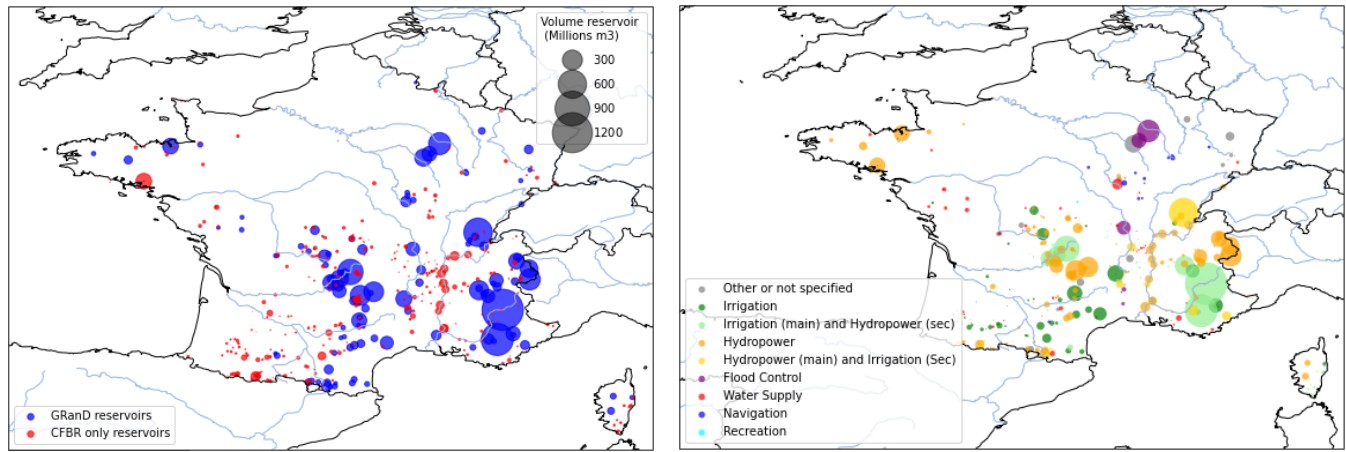

**Figure B1.** Location and main uses of the reservoirs in the final database

### B2 Hydropower plants

We use hydropower plants data from the EU Joint Research Center (JRC) Hydro-power plants database. This database gathers
geographical coordinates, installed power capacity, plant type (run-of-river, reservoir or pumped-storage) and hydraulic head
for 4186 European plants (for a total installed capacity of 161 $GW$). 153 of these plants are located in France, representing
20.6 $GW$.

Other available datasets of French hydropower plants are the national registers of electricity generation and storage facili-
ties published annually (ODRÉ, 2016, 2018). The 2016 register gathers data from 414 hydropower plants for a total installed
capacity of 23.4 $GW$. However, as these registers do not provide the geographical coordinates of the plants, we chose to use
the JRC database. Nevertheless, we use data from the RTE 2016 national register to correct head information and classify the



plants in the 4 categories used by the French operator: run-of-river, poundage (intermediate category between run-of-river and reservoir), reservoir and pumped-storage. Fig. B2 shows the locations of the different plants included in out final database while Table B1 summarizes the differences between the databases in terms of installed capacities. Its last line details the main

features of the final database we use for this study.

| | Total | RoR | Poundage | Res. | PHS (prod) | PHS (pump) |
|---|---|---|---|---|---|---|
| National Register 2016 (ODRÉ, 2016) | 23.426 | 5.943 | 3.715 | 8.748 | 4.965 | 4.591 |
| JRC (initial categories) | 19.695 | 5.87 | - | 8.76 | 5.06 | 4.84 |
| Final database (JRC with RTE categories placed on HTUs) | 19.638 | 4.426 | 2.606 | 7.434 | 5.05 | 4.84 |
| (compared to ODRÉ (2016)) | (84.6%) | (74.2%) | (71.7%) | (86.0%) | (100%) | (100%) |

**Table B1.** Comparison of the different databases in terms of installed hydroelectric capacities (GW) in metropolitan France (without Corse et DOM-TOM)

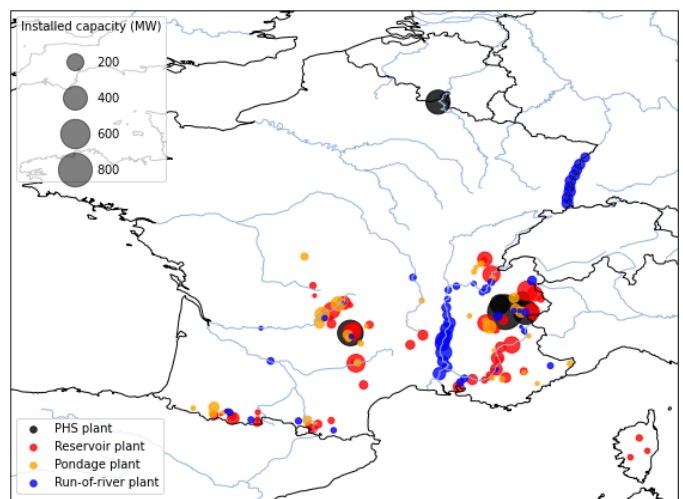

**Figure B2.** Typology of the plants in the database





## B3  Conversion factors for hydropower generation

| | Total | RoR | Poundage | Res. | PHS (prod) | PHS (pump) |
|---|---|---|---|---|---|---|
| RTE 2016 (RTE et al., 2016) | 62.6 | 31.6 | 9.4 | 15.8 | 5.8 | 6.7 |
| National Register 2016 (ODRÉ, 2016) (compared to RTE et al. (2016)) | 57.6 (92.0%) | 27.5 (87.0%) | 9.0 (95.7) | 15.6 (98.7%) | 5.8 (100%) | n.a. n.a. |
| Production plants in database 2016 (based on ODRÉ (2016)) Coefficients 2016 | 47.9 | 22.4 70.9% | 5.5 58.5% | 14.1 89.3% | 5.9 100% | n.a. n.a. |
| | | | | | | |
| RTE 2018 (RTE et al., 2018) | 66.9 | 31.3 | 10.9 | 18.8 | 5.9 | 7.4 |
| National Register 2018 (ODRÉ, 2018) (compared to RTE et al. (2018)) | 60.7 (90.7%) | 26.4 (84.3%) | 10.0 (91.7%) | 18.3 (97.3%) | 6.0 (100%) | n.a. n.a. |
| Production plants in database 2018 (based on ODRÉ (2018)) Coefficients 2018 | 48.1 | 20.5 65.5% | 6.0 55.0% | 16.2 86.1% | 5.9 100% | n.a. n.a. |
| | | | | | | |
| **Conversion factors** | | **68.2% %** | **56.8%** | **87.7%** | **100%** | |

**Table B2.** Comparison of the different available databases in terms of annual production (TWh) and calculation of conversion factors.

n.a.=not available



## Appendix C: Alternative precipitation datasets

### C1 Presentation of the datasets

#### C1.1 COMEPHORE


COMEPHORE (COmbinaison en vue de la Meilleure Estimation de la Précipitation HOraiRE) dataset provides observations of surface precipitation accumulation over metropolitan France at an hourly and kilometric resolution based on a synthesis of radar and rain gauge data. A specific processing chain has been implemented in order to address the various sources of error affecting radar data, in particular its low quality in high altitude mountainous areas like the Alps or the Pyrenees (Fumière

et al., 2020). The final database is nevertheless assumed to be the best representation of surface precipitation over metropolitan France (Fumière et al., 2020). We build a meteorologic dataset SAF_COM by replacing precipitation data in SAFRAN with data from COMEPHORE. As COMEPHORE does not distinguish solid and liquid precipitations, we keep SAFRAN's hourly ratio of solid/liquid precipitations when possible and discriminate based on the air temperature otherwise.

The differences in annual mean precipitation are generally small between SAFRAN and COMEPHORE, with an average

deviation inferior to 1.0% in COMEPHORE compared to SAFRAN (Fig. C1). However we find a small seasonal bias as this average deviation goes from -2.0% for Winter period to +1.9% in Summer. Moreover, discrepancies increase dramatically in mountainous regions, especially in the Alps and the Pyrenees. For grid points with an average elevation above 1000m, the annual mean precipitation in COMPEHORE is, on average, 10.4% lower.

#### C1.2 SPAZM

SPAZM (SPAtialisation des précipitations en Zone de Montagne) is a daily reanalysis of precipitation at the kilometer-scale, developed by EDF, the main electricity producer in France. SPAZM specifically covers the southern half of the French territory, where a large majority of hydroelectric power plants are located (Gottardi et al., 2008). Climatological precipitation outlines are first constructed based on daily precipitation observations categorized by types of oceanic circulation (weather patterns) (Garavaglia et al., 2011). These outlines are then spatially interpolated onto the kilometer-scale grid and deformed daily

according to available observations. In addition to Météo-France's observations, which are also used to construct SAFRAN, EDF's measurement network is utilized. We interpolate the daily precipitation data from SPAZM to the hourly scale and merge it with SAFRAN data to create the alternative forcing dataset SAF_SPAZM. As for SAF_COM, we keep SAFRAN's hourly ratio of solid/liquid precipitations when possible. Compared to SAFRAN, precipitations are in average 2.7% higher in SPAZM with an average bias of 7.0% in Summer, against 2.1% in Winter. Bias is heterogeneously spread over France (Fig. C1) with

bigger differences on the highest reliefs, without a clear sign (average deviation of +3.9% for grid points above 1000m).





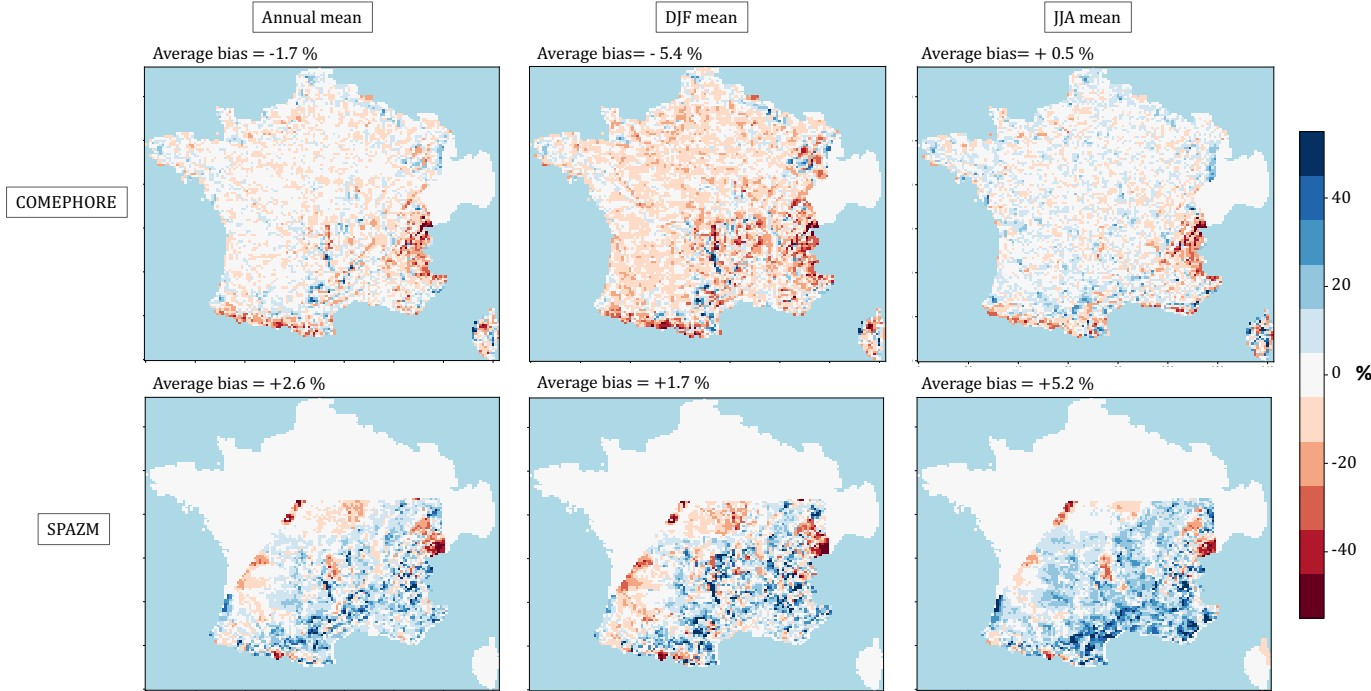

**Figure C1.** Average relative differences in total precipitation across the datasets for the period 2010-2020.

Left column: annual average bias, middle: average bias in Winter period (December-January-February), right: average in Summer period (June-July-August)

Top: COMEPHORE dataset compared to SAFRAN, Bottom: SPAZM compared to SAFRAN

## C2 Simulation of hydropower production under SAF_SPAZM

## C3 Simulation of hydropower production under SAF_COM



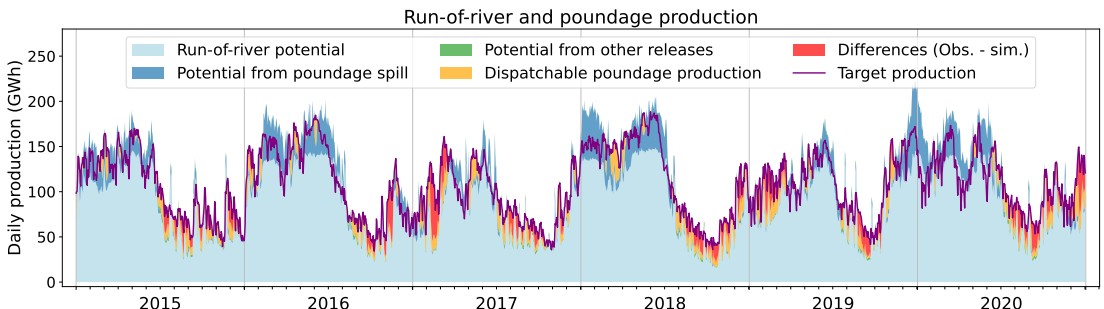

**Figure C2.** National run-of-river plant production simulated in the model when forced by SAF_SPAZM. Purple line indicates the production that has been prescribed to the model and red show the difference between this production and the one simulated in the model when forced by SAF_SPAZM. The other colors refer to the nature of the flow that contributes to the production in the model. Light blue represents the gross potential of run-of-river plants, dark blue represents the potential of spill from poundage reservoir (water overflowing from the reservoir), green represents the potential from constrained releases of poundage reservoirs and lastly orange represents the dispatchable production, generated by the water specifically released from the poundage reservoirs for power generation.

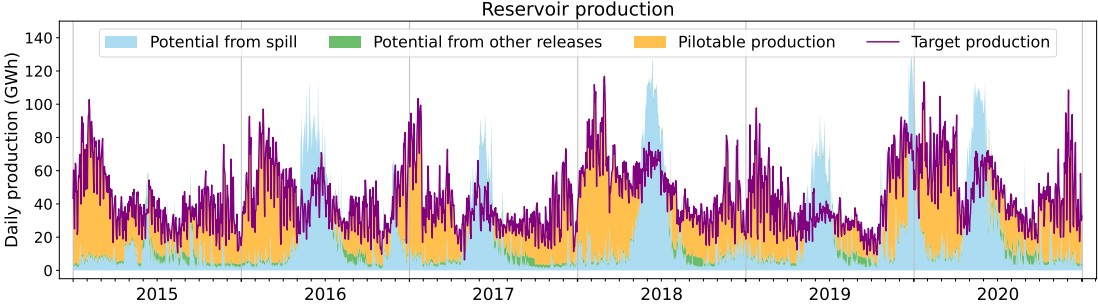

**Figure C3.** National reservoir plant production simulated in the model when forced by SAF_SPAZM

Purple line indicates the production that has been prescribed to the model.The other colors refer to the nature of the flow that contributes to this production. Blue represents the gross potential from reservoir spill (water overflowing from the reservoir), green represents the potential from constrained releases of the reservoirs and lastly orange represents the production by the water that is specifically released from the reservoir for hydropower purpose.





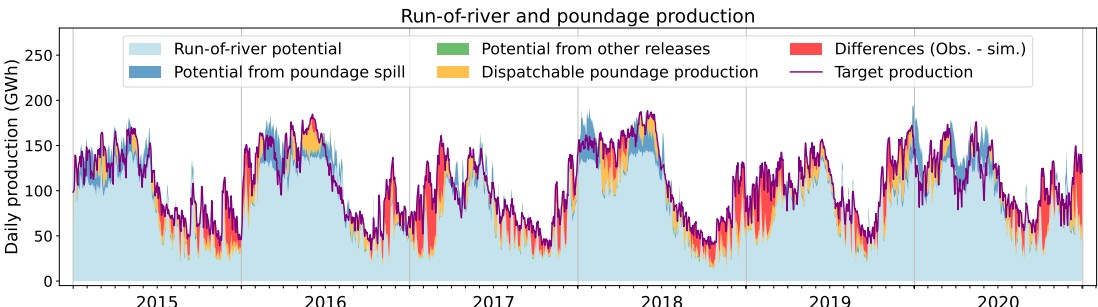

**Figure C4.** National run-of-river plant production simulated in the model when forced by SAF_COM. Purple line indicates the production that has been prescribed to the model and red show the difference between this production and the one simulated in the model when forced by SAF_COM. The other colors refer to the nature of the flow that contributes to the production in the model. Light blue represents the gross potential of run-of-river plants, dark blue represents the potential of spill from poundage reservoir (water overflowing from the reservoir), green represents the potential from constrained releases of poundage reservoirs and lastly orange represents the dispatchable production, generated by the water specifically released from the poundage reservoirs for power generation.

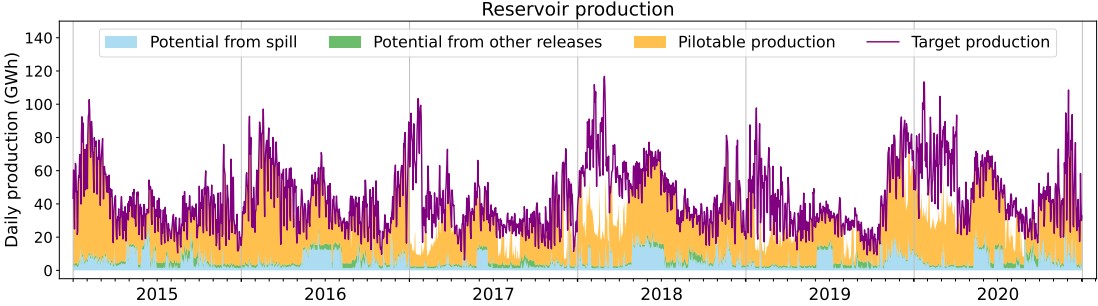

**Figure C5.** National reservoir plant production simulated in the model when forced by SAF_COM

Purple line indicates the production that has been prescribed to the model.The other colors refer to the nature of the flow that contributes to this production. Blue represents the gross potential from reservoir spill (water overflowing from the reservoir), green represents the potential from constrained releases of the reservoirs and lastly orange represents the production by the water that is specifically released from the reservoir for hydropower purpose.





## Appendix D: Hydropower network error

La Bathie power plant is the most important reservoir hydropower plant in France in terms of installed capacities. It is located in the Alps and fed by numerous water intakes represented in Fig. D1. Among them, the reservoirs of Roselend, Saint Guérin and La Gittaz as well as other intakes directly connected to rivers or glaciers.

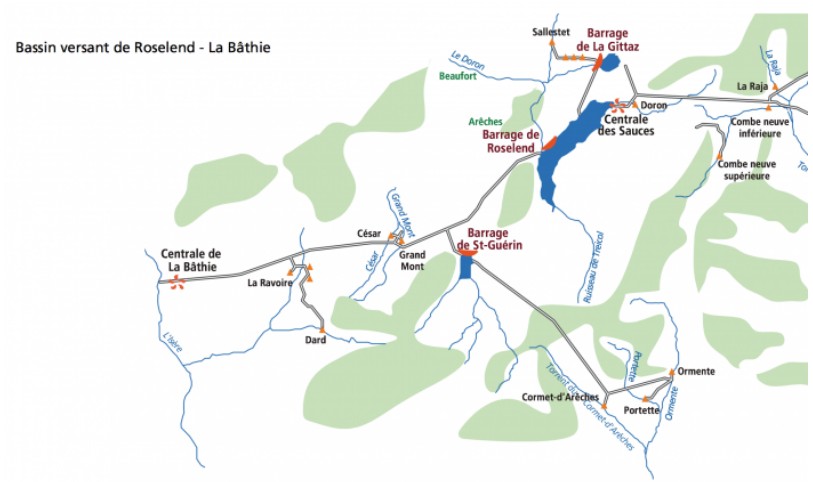

**Figure D1.** Schematic representation of the water aduction network to La Bathie power plant (source:vpah-auvergne-rhone-alpes.fr )

  Figure D2 describes the same area in HTUs space.

  We see in Fig. D2 that Roselend reservoir account for only a small part of the water being transferred to the hydropower

plant. Nevertheless, as no systematic database gathers data on hydropower plants water intakes, we keep our method to select the main reservoir as water input and then slightly moving it in order to account for the other water inputs.





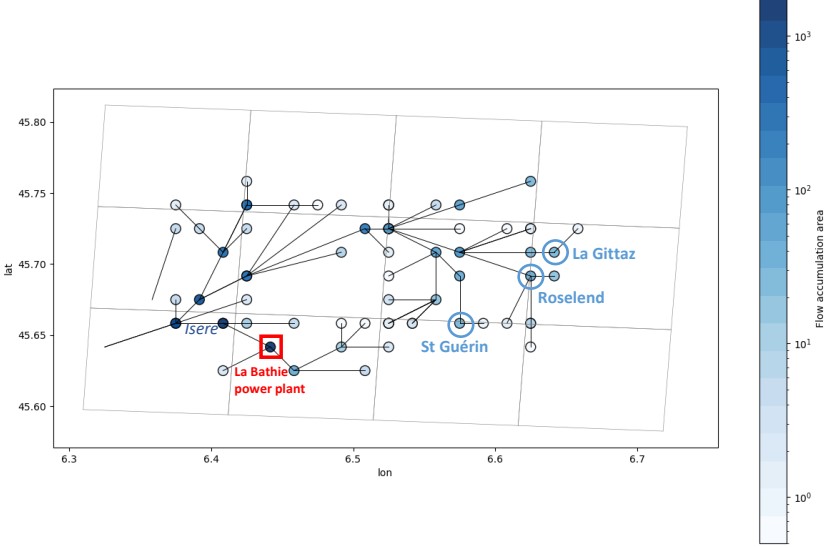

**Figure D2.** HTUs representation in the model for the same spatial area as Figure D1. The location of hydropower infrastructures is indicated.



*Author contributions.* LB developed the code, designed and executed the numerical evaluations, and wrote the first draft of the manuscript. JP, PD and PQ supervised the study. All authors jointly discussed the methodology, interpreted the results and improved the manuscript.

*Competing interests.* The authors declare that they have no conflict of interest.

*Acknowledgements.* Baratgin's PhD is supported by the French Ministry of Agriculture. The authors are grateful to EDF-DTG for providing the SPAZM precipitation data and to Meteo-France for providing SAFRAN and COMEPHORE data. HydroPortail and ODRÉ are thanked for collecting and distributing respectively the discharge and power production data. IPSL's MesoCentre is thanked for the computer time.



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
