# Peer review of "Modeling hydropower operations at the scale of a power grid: a demand-based approach"

_EGUsphere, 2023_

## Author Comment (AC1)

Response to referee comment: Anonymous Referee #1

Dear Referee 1,

Thank you for taking the time to review our paper. We are grateful for your thoughtful comments. We trust that the revisions presented below address the issues you raise.

This paper reports on the development of a model that simulates hydropower generation in France at 30-minute intervals at individual power plants, using a hydrological model used in meteorological and climatological studies. The primary boundary conditions of the model are electricity demand and meteorological information. Electricity demand is allocated to three different types of power plants: run-of-river, reservoir, and pumped-hydro-storage. River discharge and electricity generation are intensively validated and discussed.

Hydrologic models used in weather and climate studies are theoretically accurate, but often fail to achieve a simplistic objective of reproducing observed flow rates. The model constructed in this paper successfully achieved a reasonably accurate estimation of dam inflows in mountainous areas where flow reproduction is difficult. In addition, the experiment itself, in which dam releases nationwide are adjusted to match fluctuating electricity demand at 30-minute intervals, is significantly novel. At least, I have never seen or heard of such an experiment before.

Thank you for your detailed review and for acknowledging the valuable contribution that our paper provides.

While the research is excellent, the manuscript is extremely difficult to read. First, the manuscript is long. It seems the authors are trying to describe everything that was implemented and devised. The manuscript needs to be shortened, for example, by moving the treatment of exceptions (e.g., lines 180-185) to the supporting materials.

We understand your concern about the length and level of detail of the manuscript and note your comments about the difficulty of reading the manuscript.

We have made efforts to shorten the text. In particular, we have moved the paragraph you mentioned (lines 180-185) and the analysis of the simulation of the discharge of French rivers to the supplementary materials to focus the paper on the operation of hydroelectric plants in a land surface model. In addition, we plan to shorten the literature review in the introduction. Furthermore, we have changed the equations in the Methods section to make them clearer and easier to understand. Finally, we propose removing the description of the methods for PHS plants from the revised manuscript, as the results for these plants are not shown and these plants are less sensitive to hydrological conditions (at least the part of the water kept in a closed loop).

Second, the structure is not standardized. In particular, Methods and Results are written in a mixed style. Sections 2 and 3 should be put together as Methods, and Sections 4 and 5 should be put together as Results. Then, the part corresponding to introducing results in Sections 2 and 3 (e.g., Fig. 7-10) should be moved to Results, and the methodological descriptions in Sections 4 and 5 (e.g., lines 505-516) should be moved to Methods.

We agree with your suggestion to reorganize the sections to improve clarity. We have separated the Methods and Results sections more clearly. In particular, we have followed your suggestion and moved the methodological descriptions which were in section 4 to the Methods section. However, we think it makes sense to have a separate Data section. It allows to separate what belongs to the general method applicable in other regions of the world from aspects specific to the case study over France.

Finally, some technical terms seem to be not adequately defined or consistently used throughout the text (e.g., dispatch, dispatchable production, poundage, and reservoir). Further polishing is needed to enhance readability.

Response to referee comment: Anonymous Referee #1

Thanks for raising this issue, we have made a concerted effort to define each term and ensure consistency throughout the revised manuscript.

**Dispatch** refers to the process by which system operators manage the distribution of electricity production from various power sources to meet the total power demand at any given time. We have added a definition in the introduction of the revised manuscript with this sentence: "*Power dispatching involves deciding which types of power plants are activated to satisfy the total power demand, based on the cost and availability of generation resources.*".

**Dispatchable production** refers to the capability of an energy generation source to be controlled or adjusted based on demand. This term is typically used to differentiate from non-dispatchable sources, like solar or wind power, which are dependent on weather conditions and cannot be controlled. Regarding hydropower, poundage and reservoir plants are considered dispatchable as they benefit from a reservoir that can store water, while run-of-river plants are non-dispatchable.

**Poundage plants** are defined by the French Transmission System Operator (TSO) as an intermediate category between run-of-river plants and reservoir plants, as they benefit from a small storage capacity (the reservoir can be completely emptied in less than 400 hours) but are counted together with run-of-river plants in the total production time series provided by the TSO. This categorization is done by the TSO and we show the locations of the different plants in Figure B2. To clarify the term, we have introduced the poundage plants as a proper category in Section 2.1: "*Poundage plants are defined in some regions as a subcategory of reservoir plants that benefit from a limited storage capacity. As for reservoir plants, the plant and the reservoir have to be located separately*."

A **reservoir** is a natural or artificial lake behind a dam that is used to store water. The water from the reservoir can be directed to a power plant to generate electricity. The associated power plant is then called "**reservoir power plant**". As explained above, in France some such reservoir power plants are called "poundage power plants" if the reservoir can be completely emptied in less than 400 hours.

Line 166 "we thus select as feeding reservoir the one that maximizes the potential function…": Why did the authors seek the feeding reservoirs by using an algorithm? Collecting the published facts or inferring from satellite images sounds more reasonable and accurate. A bit more elaboration is needed here.

Thank you for your question, which was not sufficiently explained in our manuscript. To our knowledge, there is no dataset detailing the links between reservoirs and hydropower plants. In France, reservoirs are managed by private companies and details of their operation are considered a commercial secret. Analyzing satellite imagery could be an option, but the water pipes connecting reservoirs and power plants are often underground or on steep slopes and thus invisible to satellites.

We have included this sentence in the revised manuscript: "*Since datasets describing these connections are rarely available, we use an algorithm to infer these connections.*".

Line 180-185: I appreciate the authors providing every detail, but too much information hampers readability. Perhaps this part can be further shortened or moved to supplementary material.

Thanks for your suggestion, we have moved this part to Appendix A: Locating hydroelectric infrastructures on the river network.

Line 280 "the result of the dispatch of the total power demand": What does "dispatch" mean? A clear definition should be added.

Thank you for highlighting the need for a clear definition of "dispatch" in our manuscript. In the context of our study, "dispatch" refers to the process by which system operators manage the distribution of the electricity demand to various power sources to meet the total power needs at any given time. Dispatching involves deciding which types of power plants to activate and the level of power

production from each, based on factors such as the demand, cost, and availability of generating resources. We have added the sentence ""*Power dispatching involves deciding which types of power plants are activated to satisfy the total power demand, based on the cost and availability of generation resources.*"" in the introduction of the revised manuscript to better explain this term.

Figure 12 legend: is "bias<-50%" correct? From the text, I read "Abs(bias)>50%".

Thanks for pointing out this inconsistency between the legend in Figure 12 and the description in the text. In practice, all plants fall into one of the two cases: abs(bias)<50% or bias<-50%. "*Bias<-50%*" and "*Abs(bias)>50%*" therefore refer to the same category of plants. However, we agree that it would be clearer to use the same term and will use "*bias<-50%*" consistently in the revised manuscript.

Figure 14: First, is "poundage production" identical to reservoir production? Second, if this is the case, why must run-of-river and reservoir production be shown simultaneously (they are discussed in two different subsections)?

Thanks for raising this issue on the definition of 'poundage', which is used inconsistently throughout the manuscript. Poundage plants are an intermediate category between run-of-river plants and reservoir plants, as they benefit from a small storage capacity but are counted together with run-of-river plants in the total production time series provided by the TSO. This categorization is done by the TSO and we show the locations of the different plants in Figure B2. Since the production by poundage and run-of-river plants are combined in the "*run-of-river*" production time series published by the TSO, we use this time series to drive the operation of the reservoirs connected to poundage plants. Figure 14 details thus the contribution of pure run-of-river plants (without any reservoir) and poundage plants to satisfy the "run-of-river" production target. The reservoirs of the plants referred to as "*reservoir plants*" by the TSO are themselves operated based on a different time series, which gathers the production of reservoir plants and is presented in Figure 15. That's why poundage and reservoir production are discussed in two distinct subsections.

To clarify this issue, we have introduced the poundage plants as a proper category in Section 2.1: "*Poundage plants are defined in some regions as a subcategory of reservoir plants that benefit from a limited storage capacity. As for reservoir plants, the plant and the reservoir have to be located separately*". We have then added more details about the different operations of poundage and reservoir plants in section 3.3.2:
 "*Data of observed production for hydropower plants in the French power grid are published from 2015 by the French electricity transmission system operator RTE at a 30-minute timestep for 2 categories of plants (RTE, a):*
 - *River production $D_{river,t}$ that gathers the production of pure run-of-river power plants and poundage power plants (reservoir plants with a storage below 400h);*
 - *Reservoir production $D_{res,t}$ that gathers the production of reservoir power plants with a greater storage capacity.*
*In our model, $D_{river,t}$ is then used to drive the production of run-of-river and poundage power plants, while $D_{res,t}$ is used for the reservoir power plants with greater storage capacity. We use the classification established by RTE and illustrated in Fig.B2.*"

Third, the legend "differences" should be reconsidered because it is hard to know what variable was compared here.

We agree that the legend "*differences*" is not explicit. It refers to the differences between the target production and the simulated production.

Finally, I feel that too much information is crammed into one figure. Maybe it might be more understandable if you split it into multiple panels.

Response to referee comment: Anonymous Referee #1

Thank you for your suggestion. We propose to split the figure into two panels, the first showing the differences between simulated and observed production, while the second displays the decomposition of production across the different sources (run-of-river, spill from reservoirs of poundage plants, constrained releases from reservoirs of poundage plants and finally dispatchable production by poundage plants).

Figure 15: Is "pilotable production" identical to "dispatchable production" in Figure 14? The line for "target production" is hard to see.

Thank you for pointing out this issue. The use of "pilotable" in Figure 15 is a mistake and it has been replaced by "dispatchable" in the revised manuscript.

The line for the target production is the purple line, which shows a large intra-annual variability, as the production by reservoir plants follows the residual demand pattern.

---

## Author Comment (AC2)

Response to referee comment: Anonymous Referee #2

Dear Referee 2,

Thank you for taking the time to review our paper. We are grateful for your thoughtful comments. We trust that the revisions presented below address the issues you raise.

The authors presented a model to simulate hydropower within the routing module of land surface models with a more detailed representation of hydropower plants and their operations. Specifically, the model is validated to produce hydropower at a 30-minute time-step for individual hydropower plants of three types: run-of-river, reservoir, and pumped-hydro-storage. Such a level of detailed representation in hydropower simulation is quite impressive.

Thank you for your review and for recognizing the potentially valuable contribution that our paper provides.

However, the authors did not demonstrate the value of using such a model. For example, how could such a model be useful to dispatch hydropower in the presence of intermittent renewable resources like wind and solar, and what could be its implications for other types of storage (e.g., batteries)? Answering such questions could be a valuable scientific contribution of the paper. Instead, much of the paper demonstrated the validation of the model to replicate observed river discharge and hydropower at the power plants.

We completely agree that exploring the questions you raise would be insightful for the scientific community. However, it would require the use of a power system model able to predict the dispatch of power demand to the different power sources, both non-dispatchable (photovoltaic, wind, run-of-river) and dispatchable (nuclear, gas, hydropower reservoir, …). The operation of hydropower reservoirs depends thus both on the predicted dispatch and the water resources available, thus requiring in the end the coupling between a power system model and a hydrological model. From our perspective, our paper lays the ground for such development and analysis.

On the one hand, we have shown that our approach can realistically simulate the hydroelectric potential within a land surface or hydrological model (Figure 13), which is crucial for informing power dispatch models about the potential output of hydroelectric plants. Given the uncertainties in atmospheric forcings, land surface processes, river discharge, and the hydropower network, achieving this level of detail is not straightforward.

On the other hand, we have shown that our approach can simulate a realistic operation of individual reservoirs based on the aggregate demand for dispatchable hydropower over a power grid. Indeed, when forced to replicate the national historical production (assumed to be the demand in our case), the model operates the reservoirs similarly to what is observed in terms of stock evolution (Figure 16). This is a significant step forward toward coupling with electrical system models, as these models often represent a single aggregated power plant per electrical zone. Indeed, it allows us to explore at the individual plant level the effects of a dispatch simulated at the aggregate level.

Building on this foundational work, the natural progression is to couple our model with a power system model to explore the dispatch of hydropower in the presence of intermittent renewable resources like wind and solar. To this end, we have recently submitted a manuscript to Applied Energy, presenting a coupled modeling framework that expands on the initial methodology detailed in this paper and is applied to energy scenarios with high shares of renewable.

Response to referee comment: Anonymous Referee #2

We hope that the reformulation of the introduction will make this clearer in the revised manuscript.

Also, the authors' claim of "operations at the scale of a power grid" seems a bit misleading. The hydropower is simulated based on exogenous (observed) demand for hydropower but not based on the operation of a power grid, i.e., the model did not dispatch hydropower to meet grid-level demand considering other generation, storage, and transmission facilities.

We understand that the expression "*power grid*" can be misleading. We wanted to specify the geographic scale of our study and felt "*national*" would have been too limited as power grids are not necessarily national. Moreover, it highlights the fact that our method allows for the joint operations of all reservoirs within a given power grid in contrast to previous methods that operate each reservoir independently. We propose to clarify this in the introduction of the revised manuscript by defining it as the "*geographical scale relevant to electricity production and use*".

Moreover, the paper is not well-written. It is long and written in the form of a technical report (e.g., there are 18 figures and quite a few sections/paragraphs of a single sentence).

We acknowledge your concerns regarding the structure and length of the manuscript. We have revised the structure of the manuscript according to the suggestions of reviewer 1. In particular, we have moved the validation of river flows to the Appendix to focus more on hydropower operations, which removes 3 figures. Furthermore, we have removed small subsections (2.4.1, 2.4.2, and 2.4.3 for example) and reformulated them into a larger paragraph. Finally, we also propose to shorten the literature review presented in the introduction.

In summary, I suppose the study requires extended experiments and analysis to demonstrate the value of the proposed model, while the manuscript itself needs to be substantially improved.

We hope that the above replies to your comments clarify the value of this study and its originality relative to the current state-of-the-art.

---

## Author Response (AR2)

Dear Referee,

Thank you for taking the time to review our paper. We are grateful for your thoughtful comments.

"The revised manuscript is substantially improved, where the authors have effectively addressed my previous comments. The proposed modeling approach represents an important contribution to the literature, offering a method to simulate hydropower production with fine spatiotemporal resolution within the context of power-grid-level electricity demand. Therefore, I recommend the publication of this study in HESS.

Yet, I have a minor suggestion. The authors could add a discussion (perhaps in section '5.3 Perspectives') on how the detailed hydropower production model could be instrumental in more sustainably planning future hydropower expansion. This discussion is particularly relevant in the context of hydropower's future role under economy-wide decarbonization, which is likely to significantly increase hydropower demand (see Chowdhury et al. (2024), "Hydropower expansion in eco-sensitive river basins under global energy-economic change," https://doi.org/10.1038/s41893-023-01260-z). Including this discussion could provide valuable insights for future studies to utilize the developed model's capabilities effectively in the sustainable planning of hydropower."

Thanks for your suggestion, we have added a discussion on this point at the end of the conclusion (section 5.3): *"Such a detailed model could also be instrumental in planning future hydropower expansion more sustainably. It would help assess the demand satisfied by new hydropower plants at the grid scale, considering both existing and planned hydropower plants. Besides, the model could evaluate the potential impacts of new projects on river discharges and ecosystems."*.